# EcoFace: Audio-Visual Emotional Co-Disentanglement Speech-Driven 3D Talking Face Generation

**Jiajian Xie**[1], **Shengyu Zhang**[1]*, **Mengze Li**[1], **Chengfei Lv**[2]*, **Zhou Zhao**[1]*, **Fei Wu**[1]
[1]Zhejiang University  [2]Alibaba
{xiejiajian, sy_zhang, mengzeli, zhaozhou, feiwu}@zju.edu.cn
chengfeilv@alibaba-inc.com

## Abstract

Speech-driven 3D facial animation has attracted significant attention due to its wide range of applications in animation production and virtual reality. Recent research has explored speech-emotion disentanglement to enhance facial expressions rather than manually assigning emotions. However, this approach face issues such as feature confusion, emotions weakening and mean-face. To address these issues, we present EcoFace, a framework that (1) proposes a novel collaboration objective to provide an explicit signal for emotion representation learning from the speaker's expressive movements and produced sounds, constructing an audio-visual joint and coordinated emotion space that is independent of speech content. (2) constructs a universal facial motion distribution space determined by speech features and implement speaker-aware generation. Extensive experiments show that our method achieves more generalized and emotionally realistic talking face generation compared to previous methods [1] .

## 1 Introduction

Speech-driven 3D facial animation is an important and challenging problem that aims to generate lip synchronization and facial expressions based on arbitrary speech signals, which has several applications such as games, movies and virtual reality. Over the past few years, research has advanced from mapping speech to lip movements Cudeiro et al. (2019) to include details of facial movements Fan et al. (2022); Xing et al. (2023). However, emotional information is often subtle compared to speech content information, leading the model to focus excessively on changes in lip movements but weakly on emotional details. Recently, several studies have attempted to explicitly construct emotional features of speech, where a separate representation of speech content and emotion allows the model to focus on learning specific types of features and reduce mutual interference between them.

Manually assigning emotions is an emotion construction method used in early research, which constructs an independent expression space by encoding specified emotional features. MeshTalk (Richard et al., 2021) uses the video of the target emotion as the specified feature and EMOTE (Daněček et al., 2023) uses emotion labels and intensity labels as the specified feature. These methods allow for the construction of an emotional space that is independent of the speech content, making the generation of emotional talking faces directional. However, this is difficult to apply in automatic generation scenarios due to the need for manual input. More recently, EmoTalk (Peng et al., 2023b) employs a disentanglement of speech content from emotion, allowing the model to extract content features and emotion features from speech without manual labeling. Specifically, it exchanges emotion embeddings of two speech segments with different contents and emotions during training, separating the emotion space from the content space. However, the lack of clear supervisory signals of content and emotion is prone to the **feature confusion** problem, where the same driving result may occur for different emotional speech with the same content. Furthermore,

---

*Corresponding authors
[1]Video samples and source code are available at https://ecoface1.github.io/

due to the varying intensities of emotions and the different ways in which emotions are expressed in individual faces, the method suffers from **emotion weakening** and **mean-face** problems. To summarize, the research of speech emotion-content disentanglement is under-explored and challenged with feature confusion due to the interaction of speech content and emotional information, emotion weakening due to differences in emotion intensity, and mean-face due to differences in the speaker's style of emotional expression.

To address the above problems, we propose EcoFace, a framework that distills joint and coordinated audio-visual emotional features and enables personalized face generation. For the feature confusion and emotion weakening problem, we aim to provide explicit signals for the representation of emotional motions and emotion differentiation. Specifically, we design an audio-visual loss to supervise the discrepancy between emotional information captured by the emotional disentanglement encoder from facial motion and the audio stream, promoting synergy between audio-visual emotion features while mitigating interference from speech content. Additionally, we construct a contrastive-triplet loss to provide a discriminatory signal for different types and intensities of emotion features, facilitating the construction of a well-differentiated emotion space. As for the mean-face problem, we construct emotional motion generator (EMG), a module that first maps speech features with different contents and emotions into a low-dimensional continuous distribution space representing universal facial movement patterns, and then the speaker-aware decoder generates stylized facial action results based on stylistic encoding. Using a low-dimensional distribution space as an input to the decoder not only maintains the fixed regularity of facial action generation, but also amplifies the stylistic information, enabling the capture and generation of different styles of action details. Moreover, as well-supervised signal facilitates feature disentanglement (Ding et al., 2020), we pre-trained a lip-sync discrimination expert and constructed a visual emotion loss using emotional disentanglement encoder.

In summary, the main contributions of our work are as follows: (1) We propose an audio-visual emotional co-disentanglement method that uses explicit supervisory signals to construct an audio-visual joint and coordinated emotion space independent of speech content. (2) We implemented a mapping of emotional speech to a low-dimensional distribution space of universal facial movement and designed an emotional motion generator for stylized talking face generation. (3) Experiments show that our EcoFace outperforms other state-of-the-art baselines from the perspective of expression quality and lip synchronization metrics.

## 2 RELATED WORK

**Speech-Driven 3D Facial Animation**  Speech-driven 3D facial animation is a task to generate realistic facial animations based on speech. Early research in this domain primarily focused on mapping-based approaches, including capturing the relationship between visual and face action units (FAUs) (Ekman & Friesen, 1978) and building phoneme-viseme mapping relationships (Taylor et al., 2012; Xu et al., 2013; Edwards et al., 2016). However, all these methods require a lot of time and manpower to construct matching relationships and are difficult to generalize to new persons.

In contrast, deep-learning-based approaches have emerged as a more scalable solution, allowing models to automatically learn motion patterns from data to generate 3D facial animations. VOCA (Cudeiro et al., 2019) proposes a temporal convolutional neural network that takes speech and a silent 3D mesh template as input to generate realistic animations. FaceFormer (Fan et al., 2022) focuses on the long temporal sequences and successfully uses a transformer decoder (Vaswani, 2017) to obtain content information to generate sequential mesh sequences. CodeTalker (Xing et al., 2023) introduces VQ-VAE (Van Den Oord et al., 2017) to learn discrete motion priors to capture the speaker's movement characteristics and solve the overly smooth facial motions problem. SelfTalk (Peng et al., 2023a) further enhances generation quality through the introduction of a consistency loss. All these approaches, however, neglect the interplay between speech-driven emotions and corresponding facial expressions.

**Emotional 3D Face Animation**  Recent studies have underscored the critical role of emotion in creating realistic and expressive 3D facial animations by integrating emotional information. Specifically, MeshTalk (Richard et al., 2021) extracts emotional action potential space from the target video to synthesize emotionally detailed actions such as eyebrow movements. EMOTE (Daněček et al.,

2023) employs one-hot labels to control emotion, producing emotional facial motions through an emotion-content disentanglement loss. However, these methods require explicit control, lacking a direct connection to the emotion conveyed in the actual speech. Addressing this issue, EmoTalk (Peng et al., 2023b) disentangles speech emotion from content to generate emotional blend shapes based solely on audio input. However, EmoTalk's disentanglement process lacks a clear supervisory signal, leading to the problem of feature confusion. And it still needs to specify the intensity of the emotion explicitly to avoid emotion weakening. Unitalker (Fan et al., 2024), through a multi-head architecture that effectively leverages multiple datasets with different annotations to improve the model's generalization over emotion and synchronization. However, its mapping of speech emotion and content to the same coding space leads to the problem of weak and erroneous emotions.

In this paper, we focus on disentangling the emotion of speech from the content and reducing the confusion between speech emotion and content through the synergy of audio-visual emotional features. We also construct a low-dimensional facial action distribution space and implement style-guided 3D speaking face generation.

## 3 METHOD

Our method aims to achieve realistic and emotional 3D talking face generation driven solely by speech. Our approach employs the parametric FLAME model (Li et al., 2017) as the face representation and uses the pre-trained speech encoder Hubert (Hsu et al., 2021) to extract speech content features (Sec. 3.1). As shown in Fig. 1, the overall framework consists of two main modules: (1) Speech and video emotional disentanglement encoder (EDE), which use audio-visual co-contrastive learning to achieve the extraction of emotional features in both modalities (Sec. 3.2). (2) Emotional motion generator (EMG), which converts audio content features and emotion features into speaker-aware FLAME parameter sequences for facial imitation control (Sec. 3.3).

### 3.1 FORMULATION

As shown in Fig. 1(a), the input of EcoFace consists of the speech sequence $\boldsymbol{A}_{1:t} = (\boldsymbol{a}_1, ..., \boldsymbol{a}_t)$, where $t$ is the time of audio, each $\boldsymbol{a} \in \mathbb{R}^D$ and $D$ is the sample rate, and a speaking style vector $\boldsymbol{s} \in \mathbb{R}^n$, where $n$ is the number of speaking styles. Our model then disentangles and extracts the emotion-related latent features $\boldsymbol{E}_{1:2T} = (\boldsymbol{e}_1, ..., \boldsymbol{e}_{2T})$ and content-related latent features $\boldsymbol{C}_{1:2T} = (\boldsymbol{c}_1, ..., \boldsymbol{c}_{2T})$ from the speech sequence $\boldsymbol{A}_{1:t}$, where $T$ is the number of frames, $\boldsymbol{e} \in \mathbb{R}^{1024}$ and $\boldsymbol{c} \in \mathbb{R}^{1024}$, and combines them with the style vector $\boldsymbol{s}$ to generates a FLAME-based 3D facial animation represented by a sequence of FLAME parameters $\boldsymbol{F}_{1:T} = (\boldsymbol{f}_1, ..., \boldsymbol{f}_T)$, where $\boldsymbol{f} \in \mathbb{R}^{53}$ is FLAME expression parameters $\boldsymbol{\Psi}_t \in \mathbb{R}^{50}$ concatenated with jaw parameters $\boldsymbol{\theta}_t^{jaw} \in \mathbb{R}^3$:

$$\boldsymbol{f}_t = \left[ \boldsymbol{\Psi}_t, \boldsymbol{\theta}_t^{jaw} \right]. \tag{1}$$

**3D Face Representation** Compared to predicting mesh vertices, we use the low-dimensional FLAME (Li et al., 2017) parameter as an intermediate bridge for mesh driving, as the low-dimensional data representation not only has a faster computational speed but also simplifies the learning process. FLAME defines a mesh with $N = 5,023$ vertices and $K = 4$ joints, whose geometry can be represented using parameters $\{\boldsymbol{\beta}, \boldsymbol{\psi}, \boldsymbol{\theta}\}$, where $\boldsymbol{\beta} \in \mathbb{R}^{100}$ is the shape parameter, $\boldsymbol{\psi} \in \mathbb{R}^{50}$ is the expression parameter, and $\boldsymbol{\theta} \in \mathbb{R}^{3K+3}$ is the head pose parameter. Given a set of FLAME parameters, the 3D face mesh can be obtained with:

$$M(\boldsymbol{\beta}, \boldsymbol{\theta}, \boldsymbol{\psi}) = W\left(T_P(\boldsymbol{\beta}, \boldsymbol{\theta}, \boldsymbol{\psi}), \mathbf{J}(\boldsymbol{\beta}), \boldsymbol{\theta}, \boldsymbol{\mathcal{W}}\right), \tag{2}$$

where $T_P$ outputs vertices by combining blend shapes, the standard skinning function $W(\mathbf{T}, \mathbf{J}, \boldsymbol{\theta}, \boldsymbol{\mathcal{W}})$ rotates the vertices of $T$ around joints $J$, and $\boldsymbol{\mathcal{W}}$ performs linear smoothing. For facial animation, we predict the expression parameters $\boldsymbol{\psi}$ and the jaw components within the pose parameters $\boldsymbol{\theta}$.

**Speech Content Representation** Recently, self-supervised pre-trained speech models such as Wav2Vec2.0 (Baevski et al., 2020) and Hubert (Hsu et al., 2021) have achieved good results in content representation. Based on this, we utilize Hubert as our chosen speech content encoder for facial animation generation, as Haque & Yumak (2023) has proven it superior to Wav2Vec2.0.

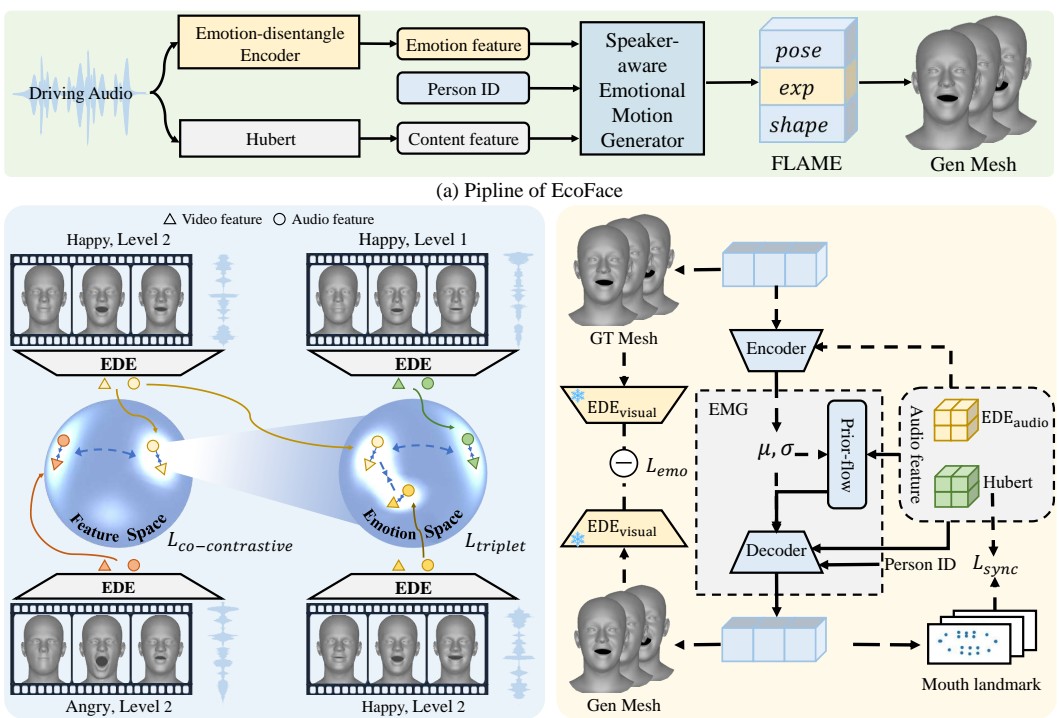

(a) Pipline of EcoFace

(b) The structure of audio-visual emotional co-disentanglement

(c) The structure of emotional motion generator

Figure 1: Overview of the proposed EcoFace. Subfigure (a) depicts the disentanglement of emotional and content features from audio, generating speaker-aware FLAME and rendering expressive meshes. Subfigure (b) illustrates how features corresponding to different emotions move away from each other in the feature space (left), while features representing different emotion intensities remain distinct within the same emotional space, and speech-video pair features are encouraged to be as similar as possible. Subfigure (c) shows the structure of the emotional motion generator. Dashed arrows indicate processes performed only during training, and only the dashed rectangle part is used during inference.

## 3.2 AUDIO-VISUAL EMOTIONAL CO-DISENTANGLEMENT

Compared with content information, speech-emotional information is often represented by subtle audio features, such as the level of intonation, speed of speech and volume. Therefore, encoding all information into the same specialized space would introduce significant uncertainty into the network, and the extraction of emotional information is disrupted by the content information. We employ the audio-visual loss described below to facilitate synergistic representations of emotional motions in speech and video, while contrastive-triplet loss is applied to provide clear separation signals for establishing an emotional space independent of content.

**Emotional Disentanglement Encoder** To match the speech content feature dimensions, we adopt Hubert's network structure as our speech emotional disentanglement encoder. For the visual emotional disentanglement encoder, since emotion is a gradual and continuous process, even in cases where there is an emotion mismatch (such as at the beginning of speech), it should still be encoded into the same emotion feature space to reduce emotional inconsistency in the generated results. Therefore, similar to Shi et al. (2022), we use an adapted ResNet-18 to extract visual emotion features at the time series level. Denoting visual feature extractor as $E_{visual}$, given a series of rendered 3D face images $\boldsymbol{P}_{1:T}$, this process can be defined as:

$$E_{visual}(\boldsymbol{P}_{1:T}) \rightarrow (\boldsymbol{v}_1, ..., \boldsymbol{v}_{2T}), \boldsymbol{v} \in \mathbb{R}^{1024}. \tag{3}$$

**Disentangled feature space construction** The overview of the audio-visual emotion co-disentanglement process is illustrated on the left side of Fig. 1(b) through the use of audio-visual co-contrastive loss. Firstly, to spatially distance the features of different emotions as much as possible, inspired by Khosla et al. (2020), we constructed emotion-supervised contrastive loss using ground

truth emotion labels. Specifically, for a set of $N$ randomly sampled audio-emotion-feature/label or visual-emotion-feature/label pairs, $\{\boldsymbol{x}_k, \boldsymbol{y}_k\}_{k=1...N}$, the corresponding batch used for training consists of $2N$ pairs, $\{\tilde{\boldsymbol{x}}_l, \tilde{\boldsymbol{y}}_l\}_{l=1...2N}$, where $\tilde{\boldsymbol{x}}_{2k}$ is the same as $\boldsymbol{x}_k$ ($k = 1...N$) and $\tilde{\boldsymbol{y}}_{2k} = \boldsymbol{y}_k$. Let $i \in I \equiv \{1...2N\}$ be the index of an arbitrary augmented sample, and let $j(i)$ be the index of other augmented sample originating from the same source sample. The loss takes the following form:

$$\mathcal{L}_{contrastive} = \sum_{i \in I} \frac{-1}{|P(i)|} \log \frac{\exp(z_i \cdot z_p / \tau)}{\sum_{a \in A(i)} \exp(z_i \cdot z_a / \tau)}, z_i = \frac{1}{2T} \sum_{t=1}^{2T} \tilde{x}_i^t, \tag{4}$$

where $z_i$ is the average of $x_i^t$ over time, since emotional features at different times under the same data should belong to the same emotion space. The $\cdot$ symbol denotes the inner dot product, $\tau \in \mathbb{R}^+$ is a scalar temperature parameter, $A(i) \equiv I \setminus \{i\}$ and $P(i) \equiv \{p \in A(i) : \tilde{y}_p = \tilde{y}_i\}$ is the set of indices of all positives in the batch distinct from $i$ and $|P(i)|$ is its cardinality. With the supervision of this loss function, features from the same category are closer together than those from different categories.

Additionally, we found that speech with different emotions corresponds to different visual representations in the periods where speech is not present, which can introduce noise into the emotional disentanglement of speech. Furthermore, as mentioned earlier, the emotions conveyed in speech are subtle and can easily be influenced by the content, leading to feature confusion. In contrast, visual emotional features are quite pronounced (e.g., the downward pressure of the corners of the mouth, eye-widening, etc.). Inspired by Wang et al. (2023); Zhang et al. (2020), we construct audio-visual loss as follows, with $z^e$ is the audio emotion feature and $z^v$ is the visual emotion feature. This loss aims to make the emotional features of speech and visual in the same sample as similar as possible, effectively sharing visual information with the speech and reducing the interference of speech content in the disentanglement process:

$$\mathcal{L}_{audio-visual} = \frac{1}{2T} \sum_{i=1}^{2T} ||z_i^e - z_i^v||^2. \tag{5}$$

In summary, the audio-visual co-contrastive loss is as follows:

$$\mathcal{L}_{co-contrastive} = \mathcal{L}_{contrastive} + \mathcal{L}_{audio-visual}. \tag{6}$$

**Single emotion space construction** As shown on the right side of Fig. 1(b), a distance threshold is necessary to prevent the issue where $\mathcal{L}_{contrastive}$ can overly compress the distance between similar features, resulting in features with the same emotion but different intensities being too close to each other. Similar to Schroff et al. (2015), we construct the emotional triplet loss that maintains a certain level of differentiation between features of different emotional levels within the same emotion type by employing the distance threshold, while also ensuring the clustering effect of features of the same emotional level. The loss is defined as follows:

$$\mathcal{L}_{triplet} = \sum_i^N \left[ ||z_i^a - z_i^p||_2^2 - ||z_i^a - z_i^n||_2^2 + \alpha \right]_+. \tag{7}$$

Given a batch with N samples, $z_i^a$ is the anchor feature, $z_i^p$ is the positive feature with the same emotion level label as the anchor, $z_i^n$ is the negative feature with the different emotion level label as the anchor and $\alpha$ is a margin that is enforced between positive and negative pairs.

## 3.3 EXPRESSIVE MOTION GENERATOR

To better capture the low-dimensional distribution space of facial motions corresponding to emotional speech and to achieve expressive and stylized 3D head motion generation, we introduce a variational autoencoder (VAE) (Kingma, 2013) to perform a generative and expressive audio-to-FLAME-parameter transformation, namely the expressive motion generator, as shown in Fig. 1(c).

**Facial motions distribution space construction** We first map the ground truth FLAME sequence to a low-dimensional distribution space. To better extract features from the input sequences and construct long-term temporal relationships in the output samples, we design the encoder and decoder as

a multilayer dilated convolutional network, where the dilation factor of the convolutional layers is gradually increased, allowing the sensory field to grow exponentially with depth. The encoder compresses the sequence of parameters of $T$ frames $\boldsymbol{f}_{1:T}$ into $T/q$ latent distributions $(\mu^{1:T/q}, \sigma^{1:T/q})$ using the speech content features $z_{1:2T}^c$ and speech emotion features $z_{1:2T}^e$ as conditions, where $q$ is the stride size of the first one-dimensional convolution in the encoder, smoothing the FLAME parameters to reduce instability between frames:

$$Enc(f_{1:T}, z_{1:2T}^c, z_{1:2T}^e) \rightarrow (\mu_{1:T/q}, \sigma_{1:T/q}). \tag{8}$$

Sampling $z_{1:T/q}^*$ from $\mathcal{N}(0, 1)$, we can get the final facial motion latent sequence:

$$z_{1:T/q} = z_{1:T/q}^* \times \sigma_{1:T/q} + \mu_{1:T/q}. \tag{9}$$

Since the Gaussian prior for vanilla VAE (Kingma, 2013) is sampled independently for each time-indexed data point, the lack of correlation can introduce instability into the sequence generation task. For this reason, we follow Ye et al. (2023) and use a flow-based prior. For the encoder output latent sequence $z_q$, the prior module, which is composed of a 1D-convolution coupling layer, a multilayer dilated convolutional network and a channel-wise flip operation, outputs the time-associated latent sequence $z_p$ conditioned on the speech content features and the speech emotion features. By calculating the KL divergence of $z_q$ and $z_p$, we ensure that the distributions of both are similar, allowing us to achieve a mapping through the speech features to the distribution of facial motions. This approach enables the production of realistic and stable results during the inference phase using the input speech features.

**Speaker-aware motion decoder**   We introduce identity embedding to guide the decoder in generating actions that capture the speaker's speaking style, such as variations in how much different speakers open their mouths in response to different emotions. The identity embedding takes a personal ID as input and generates a personalized feature $\boldsymbol{s} \in \mathbb{R}^{1024}$ consistent with the audio feature dimensions. Since for a low-dimensional distribution space, the decoder can easily sense the nuances of the input features, we simply add the personalized features to the speech features and use them as conditions, combining them with the motion latent sequences input to the decoder to generate the FLAME sequences. This process can be described as follows:

$$Dec(z_{1:T/q}, z_{1:2T}^c, z_{1:2T}^e, \boldsymbol{s}) \rightarrow \hat{f}_{1:T}. \tag{10}$$

**Training Process**   We use the Monte-Carlo ELBO loss (Ren et al., 2021) to train the EMG model. Besides, we independently train a sync-expert $D_{sync}$ that measures the possibility that the input audio content features and FLAME mouth region landmarks are in-sync. For emotion accuracy, we use the emotional disentanglement encoder trained in Sec. 3.2 to extract the emotion features of the rendered ground truth images sequence and predicted images sequence and compute their similarity. To summarize, the training loss of EMG is as follows:

$$\mathcal{L}_{EMG}(\phi, \theta, \varepsilon, \omega) = -\mathbb{E}_{q_\phi(z|f,c,e)}[\log p_\theta(\hat{f}|z, c, e)] + D_{\mathrm{KL}}(q_\phi(z|f, c, e)|p_\varepsilon(z|c, e))$$
$$-\mathbb{E}_{\hat{f} \sim p_\theta(f|z,c,e,s)}[\log D_{sync}(\hat{f})] - \mathbb{E}_{\hat{e} \sim p_\omega(e|v)}[\log \hat{e}], \tag{11}$$

where $\phi, \theta, \varepsilon, \omega$ denote the model parameters of the encoder, decoder, prior and the visual emotion encoder, respectively. $c, e, s$ denotes the condition features of EMG. $D$ is the discriminator of sync-net. The ground truth and predicted FLAME parameters are represented by $f$ and $\hat{f}$, respectively. The sequence of images rendered by $f$ denoted by $v$.

## 4   EXPERIMENT

### 4.1   EXPERIMENTAL SETUP

**Data Preparation.** To train our EDE, we use an emotional talking face video dataset, RAVDESS (Livingstone & Russo, 2018), which contains 1440 video clips of different actors speaking with 8 emotion categories. Due to RAVDESS's limited utterances, five hours of videos from the HDTF (Zhang et al., 2021), a high-fidelity talking face video dataset includes over 300 subjects and 10k

Table 1: Quantitative evaluation results. Best performance in **bold**, and the second best underlined. For better visualization, we scale up the LVE of $10^{-5}$mm.

| Methods | RAVDESS | | | | HDTF | | | VOCASET | | | MEAD | | | |
|---|---|---|---|---|---|---|---|---|---|---|---|---|---|---|
| | VE-FID↓ | LVE↓ | LSE-D↓ | LSE-C↑ | LVE↓ | LSE-D↓ | LSE-C↑ | LVE↓ | LSE-D↓ | LSE-C↑ | VE-FID↓ | LVE↓ | LSE-D↓ | LSE-C↑ |
| FaceFormer | 86.75 | 5.66 | 9.999 | 0.926 | 3.89 | 11.394 | 0.727 | 4.27 | 10.999 | 0.662 | 73.07 | 8.93 | 9.726 | 0.641 |
| CodeTalker | 82.71 | 5.57 | 9.896 | 0.942 | 3.84 | 11.755 | 0.752 | 4.11 | 10.810 | 0.667 | 73.77 | 8.25 | 10.078 | 0.646 |
| Emote | 34.01 | 3.23 | 10.452 | 0.884 | 4.07 | 11.407 | 0.736 | 3.92 | 11.174 | 0.651 | **19.36** | **4.91** | 10.049 | 0.692 |
| EmoTalk | 51.98 | 3.18 | 10.058 | 0.908 | 3.56 | 11.117 | 0.802 | 3.89 | 10.874 | 0.693 | 58.74 | 7.87 | 9.875 | 0.662 |
| UniTalker | 59.39 | 3.42 | 9.915 | 0.928 | 4.11 | 10.563 | 0.815 | 3.91 | 10.812 | 0.713 | 54.11 | 7.48 | 9.584 | 0.689 |
| GT | - | - | 10.126 | **1.014** | - | 11.535 | **0.824** | - | 10.964 | **0.813** | - | - | 10.037 | **0.715** |
| Ours | **21.57** | **2.19** | **9.616** | 1.010 | **2.61** | **10.253** | 0.823 | **3.86** | **10.757** | 0.743 | 32.44 | 5.21 | **9.113** | 0.709 |

different sentences, were used along with the RAVDESS dataset to train our EMG. A random selection of 80% of these datasets was used for training, 10% for validation, and 10% for testing. In addition, to fairly compare the models' lip-sync accuracy as well as emotion generalization, the VOCASET (Cudeiro et al., 2019) and MEAD (Wang et al., 2020) datasets will be used for the evaluation. We first processed the video in the original datasets by converting the frame rate to 25 fps and the speech sample rate to 16kHz. Then, EMOCA (Daněček et al., 2022) was used to obtain the FLAME parameters sequence.

**Compared Baselines.** We compare our model with five methods: FaceFormer (Fan et al., 2022), CodeTalker (Xing et al., 2023), Emote (Daněček et al., 2023), EmoTalk (Peng et al., 2023b) and UniTalker (Fan et al., 2024). For a fair comparison, we retrain FaceFormer and CodeTalker on the RAVDESS and HDTF datasets, since these models were trained on the unemotional dataset. As both UniTalker and EmoTalk employed the RAVDESS dataset for training purposes and EMOTE utilizes the MEAD dataset for emotion capture, we utilize their pre-trained weights. All methods require conditions on a training speaker identity during inference. Therefore, for unseen subjects in the test dataset, we obtain their predictions by conditioning on all training identities.

**Implementation Details.** We employ the Adam Optimizer across all modules. The EDE is trained for 10,000 iterations, with the batch size set to 30. This training takes about 1 hour, using a learning rate of $5 \times 10^{-5}$. Furthermore, we use 30,000 iterations with a batch size of 50 and a learning rate of $5 \times 10^{-5}$, which took about 20 hours to train our EMG. All experiments are performed on a single NVIDIA RTX 3090 GPU.

## 4.2 QUANTITATIVE EVALUATION

**Evaluation Metrics.** We evaluate our method with baselines across two factors: 1) *Facial expression quality*. Previous work used EVE (emotional vertex error) (Peng et al., 2023b), which measures the maximum $\ell_2$ error of the vertex coordinate displacement in the forehead and eye region. However, emotional expression is a coordinated process involving multiple facial muscles, including the corners of the mouth, the nose, and other facial features. In light of this, we propose the **VE-FID** (video emotion FID), an adaptation of FID (Heusel et al., 2017) that uses a video emotion feature extractor from a frame attention network (Meng et al., 2019), replacing the inception network to focus on emotion properties. 2) *Audio-Lip synchronization*. We first compute the **LVE** (lip vertex error) that is used in previous work (Richard et al., 2021). This metric computes the maximum $\ell_2$ error among all lip vertices in the test set and averages $\ell_2$ error across all frames. We then employed **LSE-D** (Lip Synchronization Error Distance) and **LSE-C** (Lip Synchronization Error Confidence) from Syncnet (Prajwal et al., 2020) to evaluate the lip movement accuracy and smoothness of the rendered faces sequence. In all the tables, ↓ indicates "the smaller the better", and ↑ indicates "the larger the better".

**Evaluation on Lip sync.** As shown in Table 1, our method achieves the highest performance in LSE-D and ranks first or second in LSE-C across all datasets. This indicates that it has strong generalization and accurate lip-synchronization capabilities. It is noteworthy that although Faceformer and Codetalker demonstrated superior lip-synchronization performance when trained on VOCASET, a decline in lip-synchronization accuracy was observed when emotion data was incorporated into the training process. This outcome further substantiates the hypothesis that our model is capable of effectively separating emotion-specific details from the speech content, thereby ensuring the precision and accuracy of the lip movements.

**Evaluation on Emotion.** Table 1 shows our method significantly outperforms VE-FID in the RAVDESS dataset, suggesting that the emotional expressiveness of our method closely resembles that of real human expressions. Although EMOTE, which utilizes ground truth emotion labels to

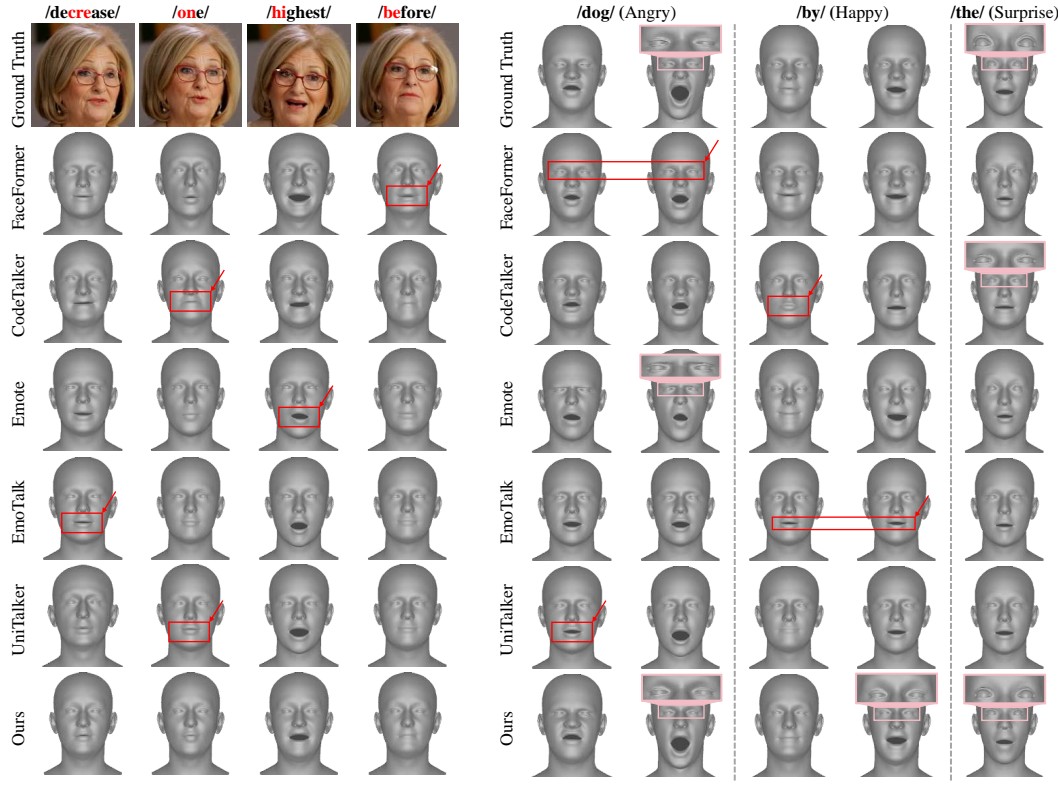

Figure 2: Comparison of generated key frame results. We show the ground truth frames used for comparison and mark the results with red arrows for out-of-sync and poor emotional quality, while zooming in on the emotional details of the eyes in the pink box.

generate expressions, achieves the best VE-FID on the MEAD dataset, our method is only second to EMOTE in this metric by a gap of 13, while EmoTalk, which uses the same training data as us, has a larger gap of 39, which demonstrates our method's generalization ability on emotion generation. Furthermore, we assess the stability of the model's emotion generation on the HDTF and VO-CASET datasets using VE-FID and compute

Table 2: Results of emotion generation stability.

| Methods | VE-FID↓ | | FID↓ | |
|---|---|---|---|---|
| | HDTF | VOCASET | Emotional | Neutral |
| FaceFormer | 61.12 | 50.05 | 40.07 | 46.08 |
| CodeTalker | 59.53 | 54.96 | 43.32 | 53.54 |
| Emote | 37.03 | 35.72 | 38.11 | 34.42 |
| EmoTalk | 27.17 | 38.22 | 38.33 | 33.73 |
| UniTalker | 37.04 | 27.59 | 39.4 | 32.779 |
| Ours | **26.43** | **23.40** | **28.54** | **31.81** |

FID metrics on the emotional and non-emotional datasets to evaluate the realism of the model's generation. Table 2 demonstrates that our model attains the optimal results on both VE-FID and FID metrics, indicating that our model is capable of stable and reliable facial emotional action generation.

## 4.3 QUALITATIVE EVALUATION

**Visual Comparison**   In Fig. 2, we compare our method with the SOTA method on the RAVDESS and HDTF test sets. As shown in subfigure (a), while most of the methods are able to generate natural lip movements without emotional interference, they are sometimes inconsistent with the ground truth. For example, when pronouncing /kr/ in 'decrease', the lips should pout. However, all methods except our method and Unitalker produce the /i:s/ pronunciation. Additionally, EMOTE shows a crooked mouth when pronouncing /'h/. As depicted in subfigure (b), the addition of emotional information results in varying degrees of incorrect lip movements across different models. For example, the lips should be pursed when pronouncing /b/, but Codetalker appeared to pout, while Emotalk seems to pronounce /ai/. Furthermore, methods other than EMOTE often yield incorrect or inexpressive expressions that do not align with the ground truth. For example, when expressing angry, Faceformer loses emotional details around the eyes, and both Emotalk and Unitalker exhibit happy

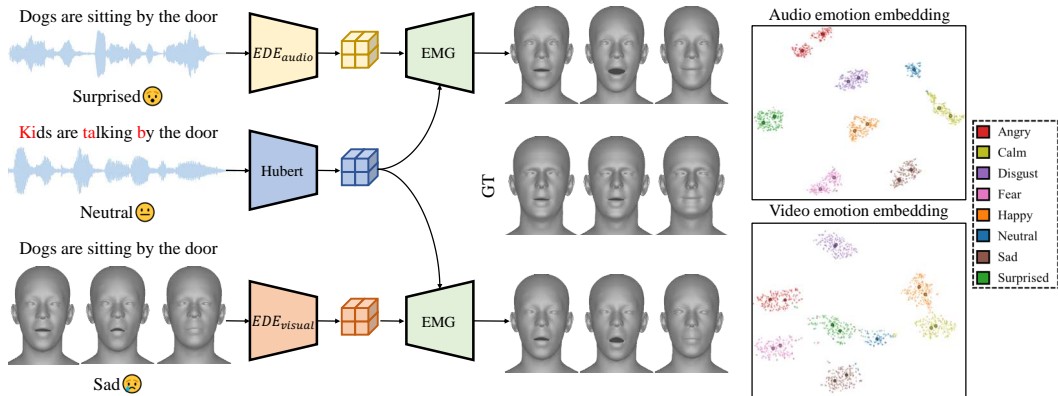

(a) Use different speech or video as emotional features     (b) TSNE visualization of feature space

Figure 3: The effect of emotion embedding. Subfigure (a) shows our audio/video emotion-driven result. Subfigure (b) shows a t-sne visualization of the distribution of our disentanglement feature space.

emotional expressions. Although EMOTE uses emotion labels to generate emotional expressions, it frequently produces unnatural expressions, such as stiff eyebrows, which deviate from the ground truth. In contrast, our model not only maintains correct lip movements but also enables detail control for different emotions. For example, in the cases of anger, happiness, and surprise, our model generates distinct representations of the details around the eyes, highlighting the impact of EDE on generating emotional expressions.

**Effect of Emotion Embedding** Fig. 3(b) demonstrates the clustering effect of our speech and video emotion embeddings, where different colors represent different emotion classes, and the centroids of different intensities within the same emotion are highlighted through stroking and bolding. It can be seen that our model effectively captures a meaningful emotion space. Additionally, to evaluate the model's effectiveness in disentangling and generating emotional expressions, we conducted an experiment: we selected a neutral speech to extract content features, which were then fed into the EMG module along with the emotional features of speech and video samples containing different emotions and speech contents. As shown in Fig. 3(a), this exchange effectively modifies the emotional expression to match the reference speech or video while maintaining precise lip-synchronization with the original speech. This shows that our EDE effectively constructs an audio-visual consistent emotionally disentangled feature space, reducing the interference of speech content with emotional information.

**User study** To evaluate the proposed model more comprehensively, we followed EmoTalk's comprehensive user questionnaire of 120 multiple-choice questions with 20 sentences selected as test cases from RAVDESS and VOCASET test datasets, and used the FLAME template to compare and analyze each of the models in terms of full-face comparison, lip-synchronization comparison

Table 3: User study results.

| Methods | full-face | lip sync | emotion expression |
|---|---|---|---|
| FaceFormer | 18.12% | 13.32% | 6.91% |
| CodeTalker | 15.53% | 11.67% | 5.32% |
| Emote | 10.13% | 13.31% | 20.89% |
| EmoTalk | 17.17% | 12.76% | 12.01% |
| UniTalker | 20.14% | 19.54% | 13.63% |
| Ours | **30.90%** | **29.40%** | **41.24%** |

and emotion expression comparison. As shown in Table 3, our model received the most positive feedback from participants, scoring the highest on all three metrics by a vote of 30.90%, 29.40% and 41.24%, respectively, compared to the other five models. It is worth noting that in terms of emotional expression, our model has an advantage over the other methods. Overall, the majority of participants considered our method superior to others.

**Ablation Studies** We conducted an ablation study to assess the contributions of different components of our model. As can be seen in Table 4, there was a modest increase in emotion expression error after removing $L_{triplet}$, with a weakening of the emotion as in Fig. 4(a), indicating its effectiveness in emotion intensity differentiation. Similarly, removing $L_{emo}$ (the last term of Equation 11)

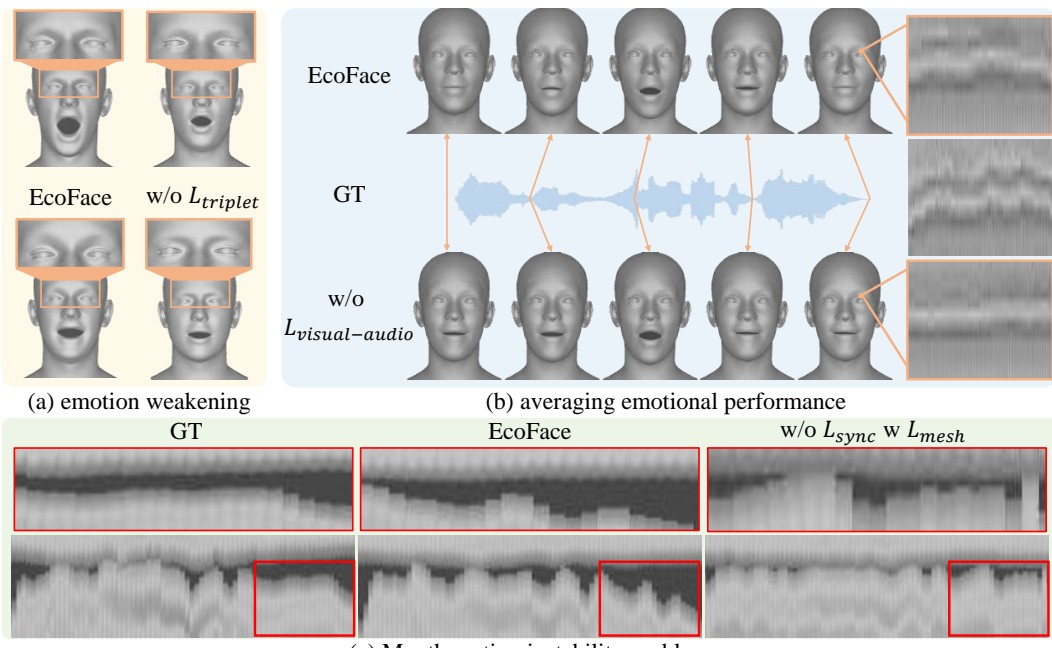

(a) emotion weakening          (b) averaging emotional performance

(c) Mouth motion instability problem

Figure 4: Ablation results while removing relevant modules.

caused a significant increase in emotion expression error, underscoring the importance of visual supervision in emotion learning and expression. Additionally, both emotion accuracy and lip synchronization declined after removing $L_{va}$, highlighting the EDE effectiveness in reducing the interference of speech content as well as lip movements on emotional features. We

Table 4: Ablation study for our components.

| Methods | VE-FID↓ | LVE↓ | LSE-D↓ | LSE-C↑ |
|---|---|---|---|---|
| w/o $L_{va}$ | 28.11 | 3.18 | 9.812 | 0.921 |
| w/o $L_{triplet}$ | 26.69 | 3.23 | 9.633 | 1.008 |
| w/o $L_{sync}$ | 22.01 | 2.22 | 10.068 | 0.946 |
| w/o $L_{emo}$ | 30.88 | 2.51 | **9.603** | **1.021** |
| Ours | **21.57** | **2.19** | 9.616 | 1.010 |

also observed that without $L_{va}$, the generated results exhibited emotion averaging. In Fig. 4(b), the left side shows generation results at different speech times. Without $L_{va}$, the results maintain a single expression over time, while EcoFace display variations in mouth corners and eye details. On the right, the eye movement sequences show that EcoFace replicates the widening and squinting movements similar to the ground truth, whereas no change occurs without $L_{va}$. This demonstrates that $L_{va}$ effectively transfers visual emotion details to speech emotion features. Finally, replacing $L_{sync}$ (the second-last term of Equation 11) with $L_{mesh}$ led to not only decreased lip synchronization but also unstable lip movements, as illustrated in Fig. 4(c).

## 5    DISCUSSION AND CONCLUSION

In this work, we propose an audio-visual emotional co-disentanglement and speaker-aware architecture for speech-driven emotional 3D facial animation. We construct an audio-visual joint and coordinated emotion space that is independent of speech content by introducing emotional motion and emotionally differentiated supervisory signals during training. At the same time, we construct a low-dimensional distribution space for facial actions to improve the decoder's sensitivity to style factors and achieve personalized generation. Extensive experiments show that our method outperforms previous methods in terms of emotional quality and lip synchronization. However, a key challenge in our model is the non-interpretability of the emotion representations, we cannot determine which disentangled emotion features correspond to specific facial changes, such as the degree of mouth opening or whether the eyes are wide open or squinting. One of the future works is to address this problem by exploring how facial muscle movements, such as changes in facial action units (FAUs), correspond to these emotion representations, to improve the precision and control of emotional expressions. Additionally, the ability to infer emotions from speech content presents important research opportunities for enhancing the accuracy of emotion generation.

ACKNOWLEDGMENTS

This work is supported by 2030 National Science and Technology Major Project (2022ZD0119100), National Natural Science Foundation of China under Grant (No. 62222211), National Natural Science Foundation of China (No. 62402429), Key Research and Development Program of Zhejiang Province (No. 2025C01026, 2024C03270), ZJU Kunpeng&Ascend Center of Excellence, Zhejiang University Education Foundation Qizhen Scholar Foundation and Ningbo Yongjiang Talent Introduction Programme (2023A-397-G). In addition, this work was supported by Alibaba Group through Alibaba Research Intern Program and Alibaba Innovation Research (AIR) Program.

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

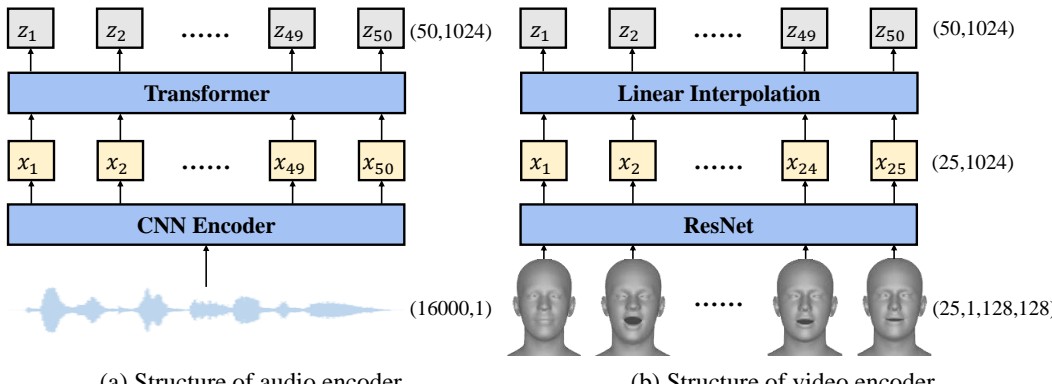

(a) Structure of audio encoder        (b) Structure of video encoder

Figure 5: Structure of the emotional disentanglement encoder.

## A  DETAILS OF MODELS

### A.1  EMOTIONAL DISENTANGLEMENT ENCODER

The Emotional Disentanglement Encoder (EDE) comprises an audio encoder and a video encoder. The architecture of the audio encoder, as depicted in Fig. 5(a), consists of a convolutional encoder followed by a transformer module. For a one-second speech segment, sampled at 16 kHz, the input to the audio encoder is a matrix of dimensions $(16000, 1)$. This is processed to yield a feature matrix of size $(50, 1024)$. The video encoder, shown in Fig. 5(b), employs a modified ResNet-18 architecture. Specifically, the first convolutional layer in the ResNet-18 is replaced with a 3D convolutional layer with a kernel size of $5 \times 7 \times 7$. The resulting visual feature tensor is subsequently flattened into a one-dimensional vector via a 2D average pooling layer. For a one-second video sequence, the input to the video encoder is a matrix of dimensions $(25, 1, 128, 128)$, where 25 represents the frames per second (fps) and each frame is a grayscale image of resolution 128 by 128 pixels. After processing by the residual neural network, a feature matrix of size $(25, 1024)$ is produced, containing feature vectors for each frame. To align the output dimensions with those of the audio encoder, linear interpolation is applied, resulting in a final feature matrix of size $(50, 1024)$.

### A.2  EMOTIONAL MOTION GENERATOR

Our emotional motion generator consists of an encoder, a decoder, and a flow prior model. The encoder, as shown in Fig. 6(a),is composed of a 1D-convolution followed by ReLU activation and layer normalization, and a condition-WaveNet. The decoder, as shown in Fig. 6(b), consists of a condition-WaveNet and a 1D transposed convolution followed by ReLU and layer normalization. The prior model, as shown in Fig. 6(c), is a normalizing flow, which is composed of a 1D-convolution coupling layer and a channel-wise flip operation. Condition-WavNet consists of multiple layers that composed of a dilated-convolution and a 1D-convolution, in which the conditional features are computed by the 1D-convolution and summed with the input features computed by the dilated-convolution to achieve conditional control. Audio emotional features and Hubert features are utilized as the audio condition of these three modules, while person ID is utilized as the person condition of the decoder.

### A.3  SYNC-EXPERT

Our sync-expert inputs a window of $T_l$ consecutive mesh landmark frames and an audio feature clip of size $T_a \times D$, where $T_l$ and $T_a$ are the lengths of the video and audio clip respectively, and $D$ is the dimension of Hubert features. The sync-expert is trained to discriminate whether the input audio and landmarks are synchronized. It consists of a landmark encoder and an audio encoder, as shown in Fig. 7, both of which are comprised of a stack of 1D-convolutions followed by batch normalization and ReLU. We use cosine-similarity with binary cross-entropy loss to train the sync-expert. Specifically, we compute cosine-similarity for the landmark embedding $l$ and audio embedding $a$ to represent the probability that the input audio-landmark pair is synchronized. The

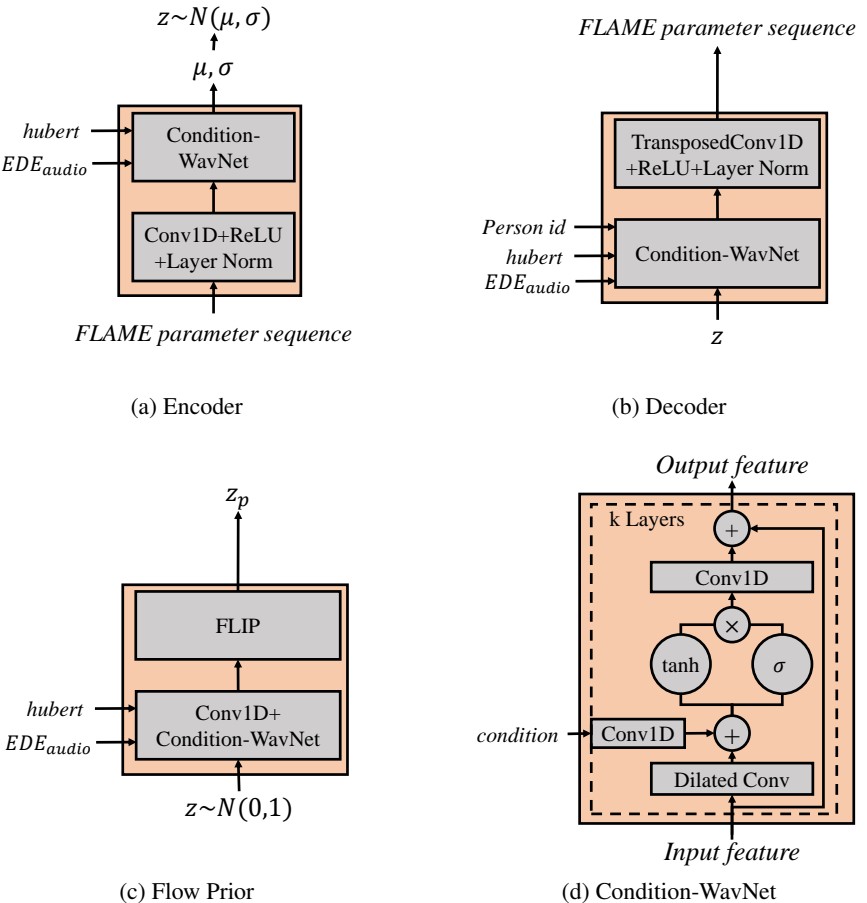

Figure 6: The structure of encoder, decoder, flow prior and condition-wavnet in emotional motion generator.

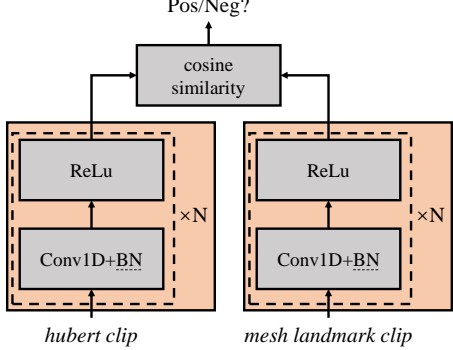

Figure 7: The structure of sync-expert.

training loss of sync-expert can be represented as:

$$\mathcal{L}_{sync} = CE(\frac{a \cdot l}{\max(||a||_2 \cdot ||l||_2, \epsilon)}) \tag{12}$$

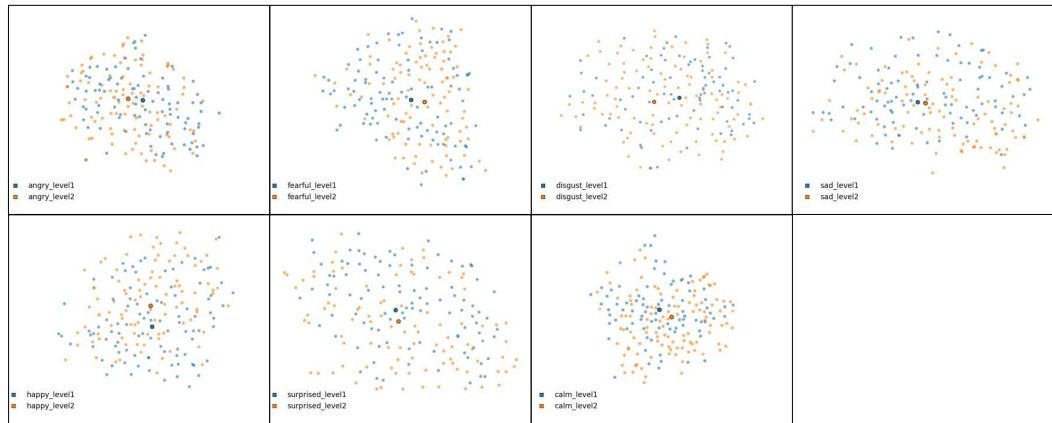

Figure 8: t-sne reault of the emotion space (w/o triplet loss).

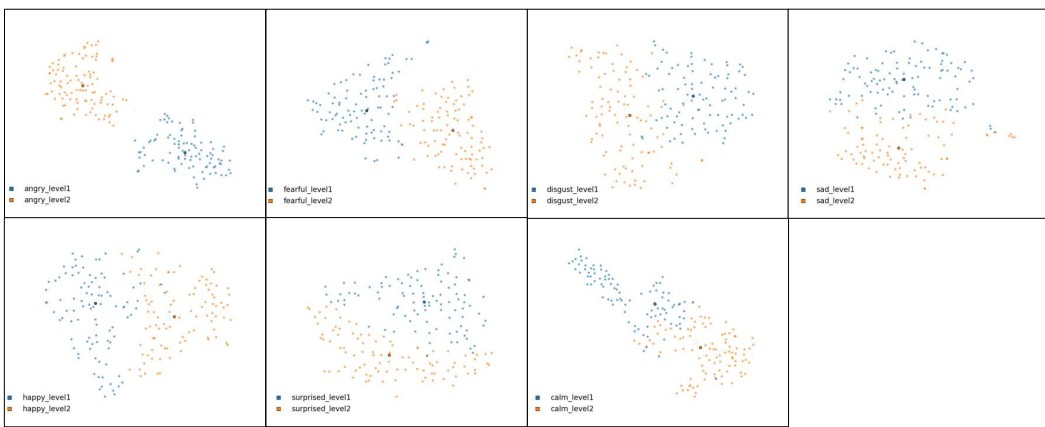

Figure 9: t-sne reault of the emotion space.

# B  VISUALIZATION OF EMOTIONAL INTENSITY DIFFERENTIATION

## B.1  THE MOTIVATION FOR COMBINING CONTRASTIVE LOSS AND TRIPLET LOSS TO DISTINGUISH EMOTIONAL INTENSITY

The Valence-Arousal model (Kensinger, 2004; Kollias & Zafeiriou, 2021) is a widely utilized framework for emotion classification. As shown in Fig.10, the VA model suggests that emotions can be represented as combinations of valence and arousal, with arousal being particularly key in distinguishing between different emotional intensities. This implies that intensity assessment is inherently tied to the emotional type assessment (via arousal), where emotions of the same type share a similar valence but can differ subtly in arousal. Therefore, our approach first allows the model to learn separate latent spaces for each emotion, effectively distinguishing the arousal and valence ranges for different emotions via the contrastive loss. Afterward, the triplet loss works within each emotion's latent space to further differentiate emotions based on their intensity by focusing on the subtle differences in arousal. This two-step process ensures that our model can effectively discriminate between both emotional types and their varying intensities.

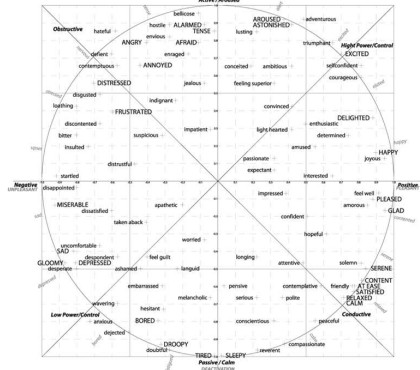

Figure 10: The Valence-Arousal model.

### B.2 T-SNE Visualization of emotional intensity

As shown in Fig. 8, we use only contrast loss to construct the emotion space, and since arousal reflects both emotion category features and emotion intensity features, and emotion intensity features are relatively homogeneous, the model ignores intensity features, and there is confusion in the distribution of emotion intensities within each separate emotion space. As shown in Fig. 9, after combining the triplet loss, there is a distinction between emotion intensity distributions within the same emotion space.

## C    Emotional Exchange Examples

### C.1    Exchange Of One Emotion

We select a news speech clip (neutral) to extract the speech content features, and select the surprised (level 2) and happy (level 1) speech of two different characters to extract the speech emotion features, and then input them into our EMG along with the content features respectively for generation. The results are shown in Fig. 12. While ensuring consistent lip synchronization, our model can generate different expression details, such as upturned corners of the mouth and wide eyes.

### C.2    Exchange Emotion In More Challenging Experiment

In order to verify whether the model will learn emotional features from speech content, we selected happy and sad emotional speech to extract speech content features, and selected related speech or video with very different emotions to extract emotional features, respectively. The generated results are shown in Fig. 13 and Fig. 14. It can be seen that our model better achieves decoupled learning of speech content and emotion.

### C.3    Exchange Of Two Emotions

We select a news speech segment (neutral) to extract speech content features, and select happy (level 2) and sad (level 1) speech of two different characters to extract speech emotion features and concatenate them together, and then input them together with content features into our EMG for generation. The results are shown in Fig. 15. While ensuring consistent lip synchronization, our model also implements smooth emotional changes, such as a gradual change in the corners of the mouth from upward to downward.

## D    Speaker-aware effects

The motivation behind designing speaker-aware generation stems from our observation that different speakers exhibit distinct speech styles and emotional expression details. If the model is trained to fit data from all speakers without considering these differences, it can lead to issues such as lip-syncing inconsistencies (e.g., pursed lips, twisted lips) and a reduction in emotional expressiveness. To demonstrate the impact of this design choice, we first trained our model without considering speaker-aware conditions and evaluated it on the test sets of the RAVDESS and HDTF datasets. The results, as shown in Table 5, indicate a slight

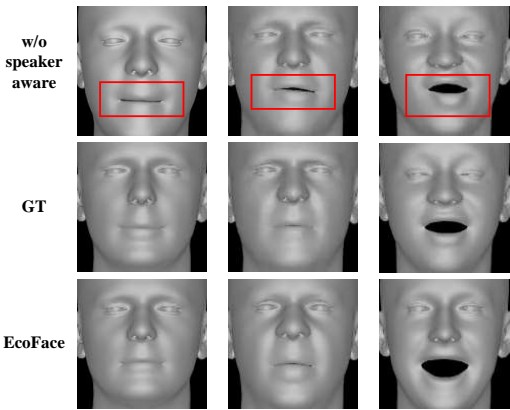

Figure 11: The Speaker-aware effects.

weakening in emotional performance, but more notably, there is a significant reduction in lip-sync accuracy. As shown in Fig. 11, the crooked, abnormally pursed lips that occur without consideration of the particular speaker can be effectively avoided by consideration of the particular speaker's style.

Table 5: Quantitative evaluation results on speaker-aware module.

| Methods | RAVDESS | | | | HDTF | | |
|---|---|---|---|---|---|---|---|
| | VE-FID ↓ | LVE ↓ | LSE-D ↓ | LSE-C ↑ | LVE ↓ | LSE-D ↓ | LSE-C ↑ |
| w/o speaker-aware | 25.67 | 3.53 | 10.552 | 0.801 | 3.47 | 11.891 | 0.711 |
| Ours | **21.57** | **2.19** | **9.616** | **1.010** | **2.61** | **10.253** | **0.823** |

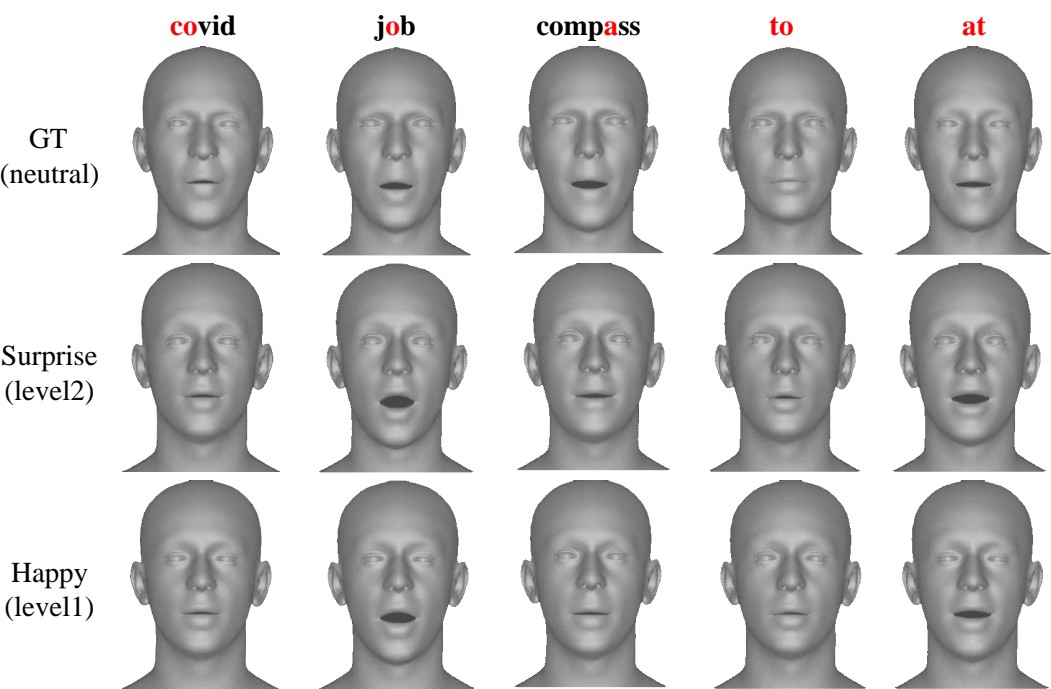

Figure 12: Generation results for exchanging whole speech emotional features.

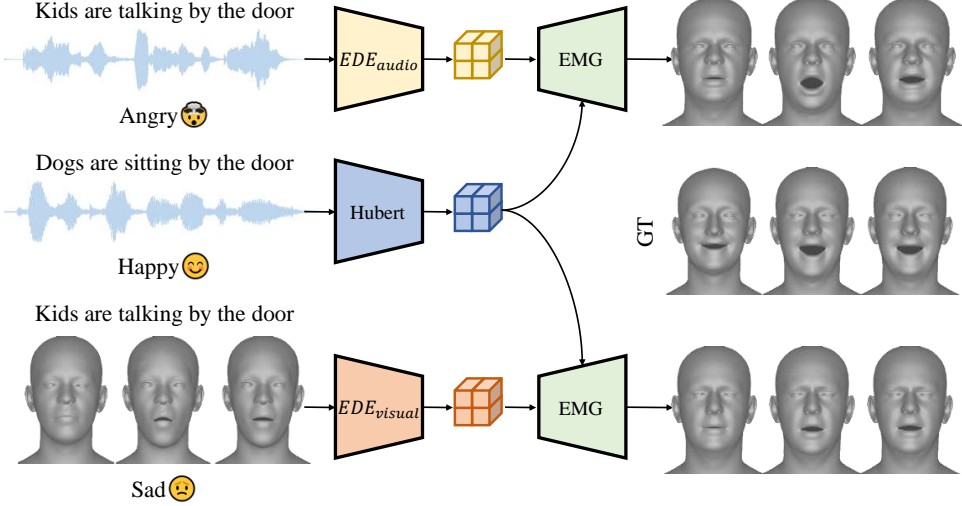

Figure 13: Generation results for exchanging emotion from positive to negative.

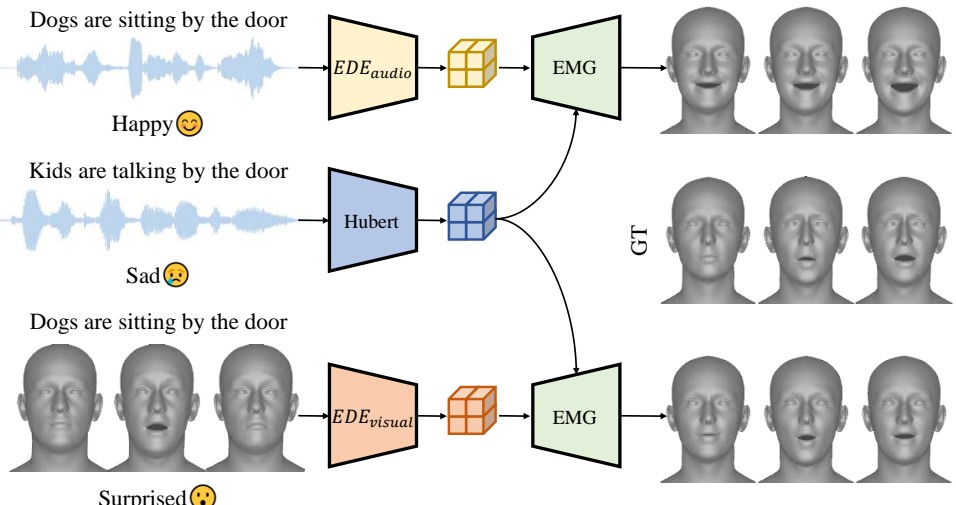

Figure 14: Generation results for exchanging emotion from negative to positive.

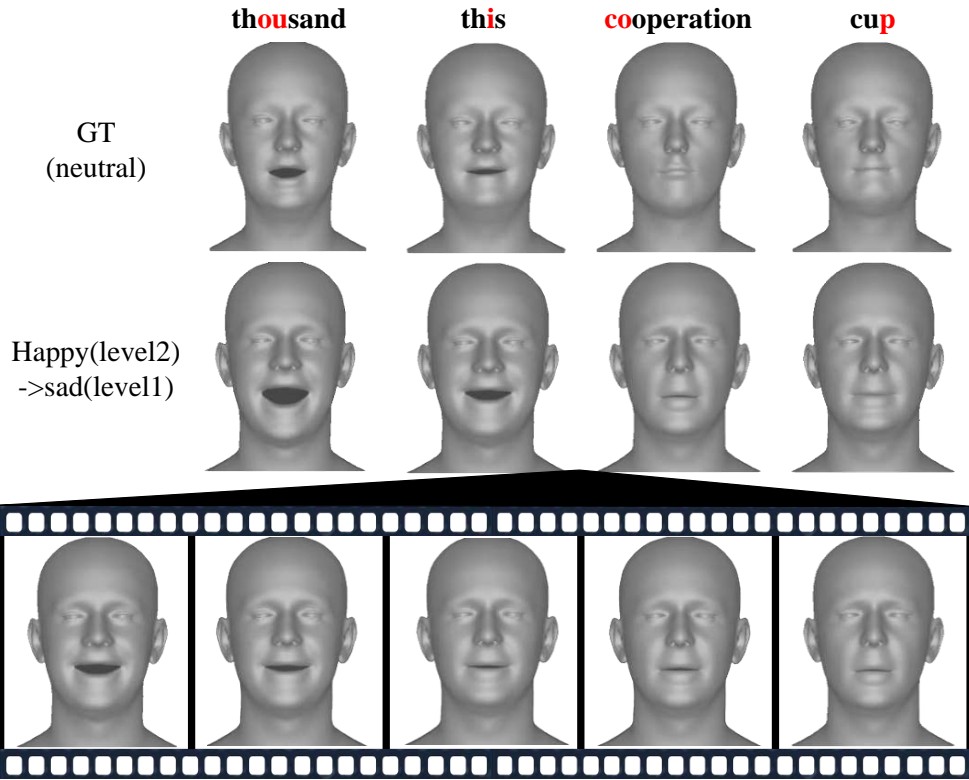

Figure 15: Generation results for exchanging two speech emotional features.

