# OpenReview forum: "EcoFace: Audio-Visual Emotional Co-Disentanglement Speech-Driven 3D Talking Face Generation"
_ICLR.cc/2025/Conference — ICLR 2025 Poster_

### Official Review · Reviewer_puML · 2024-11-03

**Soundness:** 3
**Presentation:** 3
**Contribution:** 3
**Rating:** 8
**Confidence:** 4

**Summary:**

This paper describes a framework for disentangling facial expressions of emotion from facial motions related specifically to speech.  The goal is to later re-synthesize expressive talking faces driven by speech.  Furthermore, a triplet loss is included to ensure that different expressions can be separated in the representation, and that the differences in the magnitude of same expression can be disambiguated.  To account for differences in the way that different people display facial expressions, the generator is conditioned on a learned identity embedding..

**Strengths:**

The problem being tackled is important and challenging.  We are highly sensitive to discrepancies in generated facial motion that accompanies speech.  This work tackles the problem by disentangling facial motion related to emotion from facial motion related to speech.  Furthermore, accounting specifically for the degree of expression is effective.

The work uses open data to aid re-producibility.  Furthermore, code will be made available.

The approach is effective.  The performance against the baselines is good.  The demo videos are impressive.  Talking head generators often cannot generalize to things like singing (because the sustained gestures look unnatural) but the approach here does an excellent job.  Furthermore, the differences in the articulation for different languages is very well captured.  For example, the lip-rounding in articulation of French speech is very impressive.

The ablations show the importance of the design of the components of the system.

I appreciate the inclusion and attention to detail in Section 4.3.

**Weaknesses:**

A limitation of the objective metrics, such as LVE, is that they do not account for type of error.  A larger LVE in the articulation of, say, /k/ might be insignificant, but a large error in the articulation of, say, /b/ is highly problematic.  I realize these are standard metrics, but I still see these as problematic for this reason.

See the questions/suggestions below.

**Questions:**

How is the level of emotion in the training data assigned?

On line 135 — is an element of s \in R^{n} the probability of that expression being present?

In Equation (7) how is alpha set?

In the paragraph following Equation (7), should z_l} be z_{i} to match the equation?

In Equation (11), does each term contribute equally to the loss?  There is no weighting to account for things like differences in scale?

On line 477 — you mentioned the margin that you beat the baselines by being 30.9%, 29.4%, and 41.24%.  This is not a margin though is it?  Are these not the scores that your approach attained?  The margin would be the difference.

*Suggestions*
Change references to. “speaker-specific” to “speaker-aware”.  Speaker-specific typically means that you train a different model for each individual speaker.

On line 135 you mention “where each a \in R^{D} has D sampled audio.  I think you mean that each a \in R^{D} is a D-dimensional feature vector.  I am not sure what ”D sampled audio“ means.

In Section 3.1 it would help to specify that the speech features are latent features and not typical speech features, e.g., from a Mel-filter bank.

In Section 3.1 it was not clear that for T frames of audio, why there are 2T frames of emotion-related and 2T frames of content-related features.  Maybe explain this when these are introduced.

The equations should flow as part of the sentences.  Throughout the paper you end a sentence before providing the equation.

Line 146 — replace low-latitude with low-dimensional.

On line 232, you refer to “the voiceless phase”.  Voiceless has specific meaning in speech.  To be clear here you should avoid using this terms and refer to “periods where speech is not present”.  (For reference, voiceless speech is speech produced when the vocal chords are apart.).

---

> ### Author Response · Authors · 2024-11-24
>
> Dear reviewer, we sincerely appreciate your careful and accurate understanding of our approach. We hope our response fully resolves your concerns.
> >Q1: A limitation of the objective metrics, such as LVE, is that they do not account for type of error. A larger LVE in the articulation of, say, /k/ might be insignificant, but a large error in the articulation of, say, /b/ is highly problematic. I realize these are standard metrics, but I still see these as problematic for this reason.
>
> A1: We agree with your insightful comment regarding the limitation of objective metrics like LVE. We chose to use LVE because it is a widely accepted evaluation standard. However, we also recognize the issue you pointed out: LVE does not account for the type of error. Specifically, while LVE may show a large error in the articulation of /k/ due to the mouth's opening being different from the ground truth, this may not necessarily indicate an error in pronunciation. In contrast, LSE-C and LSE-D address this issue more effectively, as they assess the matching of continuous lip movements to the speech in the time domain. These metrics can distinguish cases where a large LVE might be insignificant, like with the articulation of /k/, from cases where a large error is truly indicative of a pronunciation mistake, such as with the articulation of /b/ where the lips are closed.
>
> When both LVE values are large, it can lead to misjudgment, but using LSE-C and LSE-D allows us to avoid such situations. We also see potential improvements for this metric in mesh-driven evaluations. Since the SyncNet expert used in Wav2Lip is trained with both audio and image inputs, a possible next step would be to train a specialized SyncNet expert using 3D lip vertex from the mesh and audio as inputs. This approach, if applied to large-scale datasets, could significantly enhance the evaluation of mesh-based talking faces.
>
> >Q2: How is the level of emotion in the training data assigned?
>
> A2: In the RAVDESS dataset, the emotion intensity is divided into two categories. Therefore, we assign emotion intensity labels of 0 and 1 to the training data. These labels are primarily used in the calculation of triplet loss, where we select samples with the same intensity label as the anchor as positives, and samples with different intensity labels as negatives.
>
> >Q3: On line 135 — is an element of s \in R^{n} the probability of that expression being present?
>
> A3: In line 135, s is a one-hot embedding vector. Assuming there are n speakers, s is a vector of length n containing only 0 and 1. If we want to condition the model on the speaking style of the i-th person (where 0≤i<n), then S_i=1 and all other elements of s are set to 0.
>
> >Q4: In Equation (7) how is alpha set?
>
> A4: In Equation (7), we set α to 0.5. This choice was made after exploring alternative values, such as α=0.25 and
> α=0.7. When α=0.25, positive and negative samples become too close, reducing the model's ability to effectively separate subtle differences in emotion intensity. On the other hand, with α=0.7, the final loss converges to a larger value compared to α=0.5. Furthermore, t-SNE visualizations of the encoded feature space revealed that the separation of emotion intensities was similar for both α=0.5 and α=0.7. This indicates that at α=0.5, the model has already achieved an upper limit in separating different emotion intensities, and further increasing α does not yield additional benefits. We have included t-SNE visualizations in the appendix, illustrating the distribution of features for different emotion intensities in a single emotion space. These plots demonstrate that the features are well-separated, with distinct clusters for the two intensities within the same emotion class under the α=0.5 setting.
>
> >Q5: In the paragraph following Equation (7), should z_l} be z_{i} to match the equation?
>
> A5: Thank you for pointing out this issue. You are correct, and I have revised the notation in the paragraph following Equation (7) to use z_i, ensuring consistency with the equation.

---

> > ### Author Response · Authors · 2024-11-24
> >
> > >Q6: In Equation (11), does each term contribute equally to the loss? There is no weighting to account for things like differences in scale?
> >
> > A6: In Equation (11), since all coefficients are set to 1, each term contributes equally to the loss. This choice is made because our main research focus is on evaluating the effectiveness of emotion disentanglement rather than optimizing the performance of the generative module. Furthermore, our goal is to ensure that the model performs well in its basic configuration without relying on hyperparameter tuning. By assigning the same weights to all terms, we can assess the basic capabilities of the model.
> >
> > We also did experiment with different weightings to explore potential performance improvements. For instance, when the weight of L_emo (the last term in Equation 11) was increased to 2 while keeping the other terms at 1, the VE-FID metric on the RAVDESS dataset improved slightly from **21.57** to **21.08**. However, increasing the weight to 3 led to minimal further improvement in VE-FID (21.07), but it adversely affected lip synchronization metrics, with LSE-D increasing from 9.616 to 10.212 and LSE-C decreasing from 1.010 to 0.996. This suggests that overemphasizing the emotion term can compromise lip synchronization accuracy. Similarly, when we adjusted the weight of L_sync (the second-to-last term in Equation 11) to 2 or 3, we observed negligible changes in LSE-D and LSE-C metrics. It is evident that our model is not sensitive to the loss coefficients
> >
> > >Q7: On line 477 — you mentioned the margin that you beat the baselines by being 30.9%, 29.4%, and 41.24%. This is not a margin though is it? Are these not the scores that your approach attained? The margin would be the difference.
> >
> > A7: Thank you for pointing this out. What I intended to convey was the proportion of votes our model received relative to the total votes, and we have updated the wording in the paper to reflect this more accurately.
> >
> > >Q8: On line 135 you mention “where each a \in R^{D} has D sampled audio. I think you mean that each a \in R^{D} is a D-dimensional feature vector. I am not sure what ”D sampled audio“ means.
> >
> > A8: Thank you for pointing this out. To clarify, in this context, D refers to the sampling rate of the audio. In our experiments, the audio is sampled at 16 kHz, meaning 16,000 audio samples are collected per second. Each sample corresponds to the amplitude of the audio signal at a specific point in time. Subsequently, we flatten the audio along the time dimension into a matrix of shape (time×16000,1), from which we extract both content and emotional features.
> >
> > >Q9: In Section 3.1 it was not clear that for T frames of audio, why there are 2T frames of emotion-related and 2T frames of content-related features. Maybe explain this when these are introduced.
> >
> > A9: Thank you for pointing this out. To clarify, the feature lengths mentioned here are measured in terms of frame count. For instance, if a video has T frames at a frame rate of 25 fps, then both the emotion-related and content-related features will have lengths of 2T. It is important to note that the T referred to here differs from the T described in Q9, where T represents the duration in time. This distinction was not adequately explained in the original text, and we appreciate your feedback in pointing out this potential confusion. We have revised the relevant part in the paper.
> >
> >
> > >Q10: Suggestions Change references to “speaker-specific” to “speaker-aware”. & In Section 3.1 specify the speech features are latent features. & The equations should flow as part of the sentences. & replace low-latitude with low-dimensional. & avoid using “the voiceless phase”.
> >
> > A10: Thank you for pointing these out, we have fixed the errors in the revised version.

---

> > > ### Comment · Reviewer_puML · 2024-11-26
> > > **Thanks for addressing my questions**
> > >
> > > Thank you authors for addressing my questions.  I will maintain my score.

---

> > > > ### Author Response · Authors · 2024-11-27
> > > > **Thanks for your acknowledgment of our rebuttal**
> > > >
> > > > Dear Reviewer puML,
> > > >
> > > > Thanks for your acknowledgment of our rebuttal!
> > > >
> > > > Again, we'd like to thank you for your expertise and engaged discussion that improved the soundness and completeness of the paper.

---

### Official Review · Reviewer_kKfU · 2024-11-03

**Soundness:** 3
**Presentation:** 3
**Contribution:** 2
**Rating:** 6
**Confidence:** 4

**Summary:**

This paper addresses the challenge of separating emotional information from speech content in facial animation, which often leads to feature confusion and emotion weakening. This work, EcoFace, introduces audio-visual co-disentanglement and a speaker-specific motion generator to address these issues. By using an audio-visual loss for emotion consistency and a contrastive-triplet loss for distinct emotional space, EcoFace creates a low-dimensional motion distribution that captures speaker-specific styles for personalized animation.

**Strengths:**

1. The paper proposes a novel framework to distanglment emotion and speech content in the face synthesis field.
2. This work is well-written and well-organized.
3. The results shows the efficiency of the proposed framework.

**Weaknesses:**

Major Concerns:

1. LSE-C Range Consistency
   There appears to be a discrepancy in the LSE-C values reported in this manuscript compared to those commonly cited in the community. For instance, DINet [https://arxiv.org/pdf/2303.03988] lists an HDTF LSE-C value (as GT) of 8.9931, while the value reported in this work is 0.824. Clarification regarding this disparity is needed, as it may suggest a difference in evaluation methodology or algorithm implementation.

2. Definition Clarity for L_emo and L_sync in Ablation Study.
   The explanation of L_emo and L_sync in Table 4 is missing, which decreases in interpreting the results and understanding the contributions of each component.

3. Validity and Interpretation of VE-FID Metric
   Introducing VE-FID as a new metric to measure emotion expression is a great try. However, some results in Table 1 and Table 4 raise questions. For example, the comparison between EcoFace (21.57) and EmoTalk (51.98) shows a large difference, even when EcoFace lacks L_emo (30.88). Does this suggest that EcoFace may outperform EmoTalk on emotion-related metrics even without the emotion-related modules or guidance?

4. Lack of Results for Speaker-Specific Generation
   One of key contributions is implementing speaker-specific generation; however, the absence of results supporting this feature weakens its impact. Providing empirical evidence or case studies to demonstrate speaker-specific generation’s effectiveness would strengthen the claim.

Minor Issues:

1. On Page 3, Lines 160-161, please use “Wav2Vec 2.0” instead of “Wav2Vec2”.

2. In Figure 1(a) on Page 4, the method name is currently missing.

3. On Page 10, in Figure 4(a) and Figure 4(c), Line 518, the label/term should be corrected from “EcoTalk” to “EcoFace.”

4. Demographic Information in User Study

**Questions:**

1. What is the demographic information in the user study?

---

> ### Author Response · Authors · 2024-11-24
>
> We are grateful for your positive review and valuable comments, and we hope our response fully resolves your concerns.
> >Q1: LSE-C Range Consistency There appears to be a discrepancy in the LSE-C values reported in this manuscript compared to those commonly cited in the community. For instance, DINet [https://arxiv.org/pdf/2303.03988] lists an HDTF LSE-C value (as GT) of 8.9931, while the value reported in this work is 0.824. Clarification regarding this disparity is needed, as it may suggest a difference in evaluation methodology or algorithm implementation.
>
> A1: Thank you for pointing out the discrepancy in the LSE-C values. As mentioned in several recent works such as FaceTalk and DeepTalk, the LSE-C and LSE-D metrics have been increasingly used to measure lip synchronization accuracy in 3D mesh representations (e.g. FaceTalk[1], DeepTalk[2]). Follow these studies, we adopted the same metrics to evaluate our model's performance. However, as mentioned in these papers, the image rendered by 3D mesh is a grey model of the head, which is fundamentally different from a real human head image, which inherently differ from real human head images. This difference leads to variations in LSE-C values when compared to results evaluated on real human heads, such as those reported in DINet. Despite these discrepancies, the effectiveness of the LSE-C metric in evaluating lip synchronization remains consistent across these approaches.
>
> [1]Aneja S, Thies J, Dai A, et al. Facetalk: Audio-driven motion diffusion for neural parametric head models[C]//Proceedings of the IEEE/CVF Conference on Computer Vision and Pattern Recognition. 2024: 21263-21273.
>
> [2]Kim J, Cho J, Park J, et al. DEEPTalk: Dynamic Emotion Embedding for Probabilistic Speech-Driven 3D Face Animation[J]. arXiv preprint arXiv:2408.06010, 2024.
>
> >Q2: Definition Clarity for L_emo and L_sync in Ablation Study. The explanation of L_emo and L_sync in Table 4 is missing, which decreases in interpreting the results and understanding the contributions of each component.
>
> A2: Thank you for raising this important issue. L_emo is the last term of Equation 11 and L_sync is the second-last term of Equation 11. I have now revised the manuscript to clearly specify each component.
>
> >Q3: Validity and Interpretation of VE-FID Metric Introducing VE-FID as a new metric to measure emotion expression is a great try. However, some results in Table 1 and Table 4 raise questions. For example, the comparison between EcoFace (21.57) and EmoTalk (51.98) shows a large difference, even when EcoFace lacks L_emo (30.88). Does this suggest that EcoFace may outperform EmoTalk on emotion-related metrics even without the emotion-related modules or guidance?
>
> A3: Thank you for your insightful comment regarding the VE-FID metric. Although we did not use L_emo, our emotion encoder (**EDE**) already **disentangles emotion features** into an independent emotion space. This allows the EMG module to still receive clear **emotion-related signals**, albeit without the explicit emotional guidance from L_emo. Thus, while the absence of L_emo leads to a slight reduction in emotional generation quality, the well-disentangled  emotion features in our model still enable it to perform relatively better in terms of emotional expressiveness compared to EmoTalk in this specific metric. This suggests that the disentanglement of emotional features in our model contributes significantly to its improved performance even without explicit emotional guidance (L_emo).

---

> > ### Author Response · Authors · 2024-11-24
> >
> > >Q4: Lack of Results for Speaker-Specific Generation One of key contributions is implementing speaker-specific generation; however, the absence of results supporting this feature weakens its impact. Providing empirical evidence or case studies to demonstrate speaker-specific generation’s effectiveness would strengthen the claim.
> >
> > A4: Thank you for raising this important point. The motivation behind designing speaker-specific generation stems from our observation that different speakers exhibit distinct speech styles and emotional expression details. If the model is trained to fit data from all speakers without considering these differences, it can lead to issues such as lip-syncing inconsistencies (e.g., pursed lips, twisted lips) and a reduction in emotional expressiveness.
> >
> > To demonstrate the impact of this design choice, we first trained our model without considering speaker-specific conditions and evaluated it on the test sets of the RAVDESS and HDTF datasets. The results, as shown below, indicate a slight weakening in emotional performance, but more notably, there is a significant reduction in lip-sync accuracy. We have also included visual examples of such cases in the **Appendix D**, where abnormal scenarios are highlighted.
> >
> >
> > |||RAVDESS||||HDTF||
> > |:------:|:------:|:------:|:------:|:------:|:------:|:------:|:------:|
> > |Methods|VE-FID↓|LVE↓|LSE-D↓|LSE-C↑|LVE↓|LSE-D↓|LSE-C↑|
> > |Ours (w/o speaker specific)|25.68|3.53|10.552|0.801|3.47|11.891|0.711|
> > |Ours|21.57|2.19|9.616|1.010|2.61|10.253|0.823|
> >
> > >Q5: What is the demographic information in the user study?
> >
> > A5: The user study involved 20 participants, selected to ensure diversity and the rigor of the evaluation. All participants were either native English speakers or possessed advanced English proficiency, enabling accurate comprehension of speech content and emotional expression. The gender distribution was balanced, with 55% male and 45% female participants, and ages ranged from 22 to 45 years. Additionally, 30% of the participants had professional backgrounds related to computer visionl. The remaining 70% were general users from various industries such as education and IT services, ensuring that the study results reflected both expert insights and the subjective experiences of everyday users.
> >
> > >Q6: Use “Wav2Vec 2.0” instead of “Wav2Vec2”. & In Figure 1(a) on Page 4, the method name is currently missing. & On Page 10, in Figure 4(a) and Figure 4(c), Line 518, the label/term should be corrected from “EcoTalk” to “EcoFace.”
> >
> > A6: Thanks for your careful and accurate reading of our paper. We have fixed the errors in the revised version.

---

> ### Author Response · Authors · 2024-12-02
> **Hoping that our response could address your concern**
>
> Dear Reviewer kKfU,
>
> Thank you again for your time and effort in reviewing our work! We would appreciate it if you can let us know if our response has addressed your concern. As the end of the rebuttal phase is approaching, we look forward to hearing from you and remain at your disposal for any further clarification that you might require.
>
> Thanks in advance,
>
> Paper 10320 authors

---

### Official Review · Reviewer_AbM6 · 2024-11-03

**Soundness:** 3
**Presentation:** 2
**Contribution:** 2
**Rating:** 6
**Confidence:** 5

**Summary:**

EcoFace addresses issues in 3D facial animation: feature confusion (lack of clear signals for content and emotion), emotion weakening (difficulty in controlling emotion intensity), and the mean-face problem (limited control over individual speaker’s style of expression). The authors tackle these issues by introducing explicit signals for emotional motion representation and intensity control using audio-visual loss and a contrastive triplet loss to distinguish emotion intensities. They also generate speaker-specific, stylized facial animations with a lip-sync discriminator.

**Strengths:**

The work presents following strengths:
1. EcoFace introduces an audio-visual emotion disentanglement mechanism to supervise the discrepancy between emotional information captured by facial motion and the audio stream, effectively capturing information from both audio and video.
2. EcoFace controls within-emotion intensity by using an emotional triplet loss.

**Weaknesses:**

1. The GT videos on the webpage show severe issues, such as unnatural mouth closure and a jerky appearance, which don’t match the [original implementation](https://download.is.tue.mpg.de/emoca/assets/emoca_v2_comparison.mp4). Since EcoFace is trained on meshes obtained using EMOCA, the authors should clarify how their model produces improved animations compared to the GT animations on which it was trained, as shown in their demo videos.
2. The EMOTE comparison videos on the webpage don’t match the exemplar videos from other methods, such as [FaceTalk](https://youtu.be/7Jf0kawrA3Q?si=ZvxQT3FD27Eh-RDj), [3DiFACE](https://youtu.be/Mep5pAU3TPc?si=Pe93jkT_eC_pBjPZ), [EMOTE](https://download.is.tue.mpg.de/emote/EMOTE_SupMat_video.mp4) . Could the authors explain the observed differences?
3. Triplet loss in eq 7 operates within a single emotion space. Since it focuses on a single emotion, how does EcoFace ensure that emotion intensities for high arousal, high valence (e.g., surprise) are disentangled from those of high arousal, low valence (e.g., anger or fear)?
4. Since EcoFace claims to generate speaker-specific animations, it would help to provide more details on the number of identities in the training set and any animation results for unseen subjects for better evaluation.
5. Training details are unclear. With a batch size of 30, how are typical forward passes structured to use 2N pairs for computing contrastive loss in Eq. 4, and how is triplet loss computed per emotion in Eq. 7?

**Questions:**

1. L219: What kinds of augmentations were applied to the video frames and audio samples?
2. L225: The latent representations are averaged over the sequence. How is it ensured that continuous emotional information along the sequence is not lost? Could the authors provide more details on the contrastive loss in Equation 4, and clarify if other methods besides averaging were considered, such as mapping to a lower dimension with a learnable layer?
3. fig 1a: The method name is missing in the caption.
4. fig 1b: The triangle and circle shapes representing features lack labels indicating what they refer to.
5. fig 4 and Ablation Studies (pages 9,10): "Ecotalk" is used instead of "EcoFace."
6. What was the rationale for training the sync expert independently? How are the FLAME region landmarks obtained—are they input as rendered crops or as 3D vertices?

---

> ### Author Response · Authors · 2024-11-24
>
> Thank you for your review and constructive feedback. We hope our revisions fully address your concerns.
> >Q1: The GT videos on the webpage show severe issues, such as unnatural mouth closure and a jerky appearance, which don’t match the original implementation. Since EcoFace is trained on meshes obtained using EMOCA, the authors should clarify how their model produces improved animations compared to the GT animations on which it was trained, as shown in their demo videos.
>
> A1: We appreciate the reviewer’s observation. It is indeed true that the mesh estimations generated by EMOCA occasionally exhibit distortions, especially under exaggerated expressions, such as a surprise emotion with intensity level 2. However, in our demo videos, it is evident that models like FaceFormer and CodeTalker, which retrained on the RAVDESS dataset using the same EMOCA-estimated meshes, also produce smoother results.
>
> Upon further investigation, we identified certain cases in the RAVDESS dataset where estimation anomalies were more prevalent, particularly for Actor 12. When conditioning on this actor, mesh-based autoregressive methods like FaceFormer and CodeTalker occasionally generated distorted outputs due to the influence of these anomalies during training. We provide a vdeo [here](https://anonymous.4open.science/r/EcoFace-7A5F/demo/Actor12_gt_severe_probles_and_generation_compared.mp4) that illustrates a comparative analysis of GT, FaceFormer, and our generated results for Actor 12 in the validation set. It can be observed that GT meshes for this actor already exhibit noticeable distortions, and FaceFormer’s autoregressive approach is affected by these anomalies, resulting in warped or misaligned outputs. In contrast, our model effectively mitigates such issues. This is primarily because we adopt a latent **motion distribution space** with flow-based priors for training, rather than relying on a direct autoregressive method like FaceFormer.
>
> Although EMOCA occasionally introduces anomalous distortions, most of its mesh estimations are accurate, forming a distribution with a high density around well-estimated cases and a lower density for extreme outliers. Our approach leverages this property by focusing on fitting the overall motion distribution space. By incorporating flow priors, our model smooths the distribution sampling at each time step. Consequently, during inference, the sampled motion trajectories tend to align closely with the high-probability regions of the distribution, effectively avoiding anomalies like unnatural mouth closures or jerky facial movements.
>
> We hope this explanation clarifies how our model achieves improved animation quality compared to the GT meshes it was trained on.

---

> ### Author Response · Authors · 2024-11-24
>
> >Q2: The EMOTE comparison videos on the webpage don’t match the exemplar videos from other methods, such as FaceTalk, 3DiFACE, EMOTE . Could the authors explain the observed differences?
>
> A2: Thank you for bringing this issue to our attention. We acknowledge that the EMOTE comparison videos on the webpage did not perfectly align with the exemplar videos from other methods, such as FaceTalk, 3DiFACE, and EMOTE. This was a challenge noted by other users as well. We initially attempted several solutions discussed in [here](https://github.com/radekd91/inferno/issues/9) for the EMOTE model but were unable to resolve the discrepancies. However, during the rebuttal period, the authors of the EMOTE paper shared a new approach to address this issue. After implementing
> their suggested solution, we were able to successfully align the comparison results. We have since re-conducted the quantitative experiments, and the updated results are provided below.
>
> |||RAVDESS||||HDTF|||VOCASET||||MEAD||
> |:------:|:------:|:------:|:------:|:------:|:------:|:------:|:------:|:------:|:------:|:------:|:------:|:------:|:------:|:------:|
> |Methods|VE-FID↓|LVE↓|LSE-D↓|LSE-C↑|LVE↓|LSE-D↓|LSE-C↑|LVE↓|LSE-D↓|LSE-C↑|VE-FID↓|LVE↓|LSE-D↓|LSE-C↑|
> |Emote (origin)|34.01|3.23|10.452|0.884|4.07|11.407|0.736|3.92|11.174|0.651|19.96|4.91|10.049|0.692|
> |Emote (update)|31.79|3.22|10.133|0.911|3.90|10.953|0.803|3.95|11.010|0.702|18.19|4.82|9.913|0.701|
> |Ours|21.57|2.19|9.616|1.010|2.61|10.253|0.823|3.86|10.757|0.743|32.44|5.21|9.113|0.709|
>
> The updated metrics show that while EMOTE has improved in various benchmarks, our model still outperforms EMOTE.The webpage and the corresponding sections in the paper have been updated to reflect these revised results.
>
> >Q3: Triplet loss in eq 7 operates within a single emotion space. Since it focuses on a single emotion, how does EcoFace ensure that emotion intensities for high arousal, high valence (e.g., surprise) are disentangled from those of high arousal, low valence (e.g., anger or fear)?
>
> A3: Thank you for your insightful questions. Although arousal primarily reflects emotional intensity, it also intrinsically encodes features related to emotional type. This means that there is some correlation between emotion type and emotion intensity on the arousal dimension. For example, emotions such as surprise and anger may exhibit similar levels of arousal, but their differences in emotion type on the valence dimension place them in different quadrants of the VA model.
>
> Building on this idea, we employ contrastive loss to construct independent feature spaces for each emotion. This design ensures that even when emotions like surprise and anger have similar arousal levels, their **arousal features belong to distinct distributional spaces**, as illustrated in Figure 3(b) of the paper. By effectively disentangling arousal representations based on emotion type, we achieve differentiation in emotion intensity across emotion categories.
>
> Once these independent emotion-specific spaces are established, we further refine intensity distinctions within the same emotion type using triplet loss. This step allows us to model subtle variations in arousal intensity, enabling precise differentiation of emotion intensity levels.
>
> To further support this explanation, we have provided visualizations and detailed analyses of intensity differentiation in the **Appendix B.1**. We hope this clarifies the rationale behind our approach and demonstrates the effectiveness of combining contrastive and triplet loss for disentangling emotion intensities.

---

> > ### Author Response · Authors · 2024-11-24
> >
> > >Q4: Since EcoFace claims to generate speaker-specific animations, it would help to provide more details on the number of identities in the training set and any animation results for unseen subjects for better evaluation.
> >
> > A4: In our experiments, the training datasets (RAVDESS and HDTF) include 24 and 131 identities, respectively. We used 90% of the identities as training data (140 identities) and 10% as test data (unseen identities). Additionally, we randomly selected 10% of the training identities for the validation set, with the remaining data used for training.
> >
> > We focus on speaker-specific generation because related studies (e.g. FaceFormer[1], CodeTalker[2]) have found that different speakers exhibit different speaking styles, allowing the model to fit data from all speakers indiscriminately can lead to issues such as unnatural mouth movements. Therefore, we adopt a speaker-specific approach, as suggested in these articles, to address the aforementioned issues and mitigate the potential risk of emotional averaging. We have provided both quantitative and qualitative analyses in the appendix D to further justify the necessity of a speaker-specific approach.
> >
> > However, since our model is designed to be speaker-specific, it currently cannot generate speech animations that accurately capture the speaking style of unseen speakers, which is a limitation. Nonetheless, inspired by UniTalker[3], we could incorporate a pivot-identity conditioning into the training process. By doing so, for an unseen identity, we could use this pivot-specific as condition to quickly fine-tune the decoder in our EMG module, enabling it to adapt to the speaking style of the new speaker.
> >
> > [1]Fan Y, Lin Z, Saito J, et al. Faceformer: Speech-driven 3d facial animation with transformers[C]//Proceedings of the IEEE/CVF Conference on Computer Vision and Pattern Recognition. 2022: 18770-18780.
> >
> > [2]Xing J, Xia M, Zhang Y, et al. Codetalker: Speech-driven 3d facial animation with discrete motion prior[C]//Proceedings of the IEEE/CVF Conference on Computer Vision and Pattern Recognition. 2023: 12780-12790.
> >
> > [3]Fan X, Li J, Lin Z, et al. UniTalker: Scaling up Audio-Driven 3D Facial Animation through A Unified Model[J]. arXiv preprint arXiv:2408.00762, 2024.

---

> ### Author Response · Authors · 2024-11-24
>
> >Q5: Training details are unclear. With a batch size of 30, how are typical forward passes structured to use 2N pairs for computing contrastive loss in Eq. 4, and how is triplet loss computed per emotion in Eq. 7?
>
> A5: Our typical forward pass is structured as follows:
>
> **Contrastive Loss Calculation**:
>
> **1.  Feature Extraction and Averaging.** In the first step, we extract emotion features from the input (video/audio) along the time dimension. This results in a tensor of shape (batch_size, time_steps, feature_dim), where feature_dim = 1024. Since the emotion features across time steps for the same video/audio are expected to belong to the same emotion space, we take the average along the time dimension to obtain a fixed-size feature vector for each sample. This results in a tensor of shape (batch_size, feature_dim) where each sample represents the averaged emotion feature over time.
>
> **2. Constructing 2N Pairs for Contrastive Loss.** The Supervised Contrastive Loss involves comparing each sample with every other sample in the batch to form both positive and negative pairs. The loss calculation follows these key steps:
> + Masking Positive Pairs: We first create a mask to identify which samples belong to the same class, i.e., positive pairs. The mask matrix has values of 1 for pairs of samples with the same label, and 0 for pairs with different labels. This mask is of shape (batch_size, batch_size).
> + Pairwise Similarity Calculation: We compute the pairwise cosine similarities between the features of all samples in the batch. The computation is done using the dot product, and the result is scaled by the temperature parameter: similarity[i, j] = (features[i] · features[j]) / temperature
> + Numerical Stability: To ensure numerical stability, we subtract the maximum value in each row of the similarity matrix: similarity[i, j] = similarity[i, j] - max(similarity[i, :])
> + Masking Self-Comparisons: To avoid self-comparisons (i.e., comparing a sample to itself), the diagonal of the similarity matrix is set to zero: similarity[i, i] = 0 for all i
> + Log-Probability Calculation: The log-probability for each sample is computed using the softmax function applied to the similarity matrix, with the self-comparisons excluded: log_prob[i] = log(softmax(similarity[i, :])) for each sample i
> + Mean Log-Probability for Positive Pairs: We then compute the mean log-probability for the positive pairs by taking the sum of log-probabilities weighted by the mask, and dividing by the number of positive pairs for each sample: mean_log_prob_pos[i] = (sum(log_prob[i] * mask[i, :])) / (sum(mask[i, :]))
>
> **3. Contrastive Loss Calculation.** Finally, the contrastive loss is computed as follows:
> + The temperature-scaled mean log-probabilities for the positive pairs are averaged over the batch and scaled by the temperature and base temperature: loss = - (temperature / base_temperature) * mean_log_prob_pos, loss = average(loss) across all samples. This ensures that the model pulls together the features of positive pairs (samples with the same label) and pushes apart the features of negative pairs (samples with different labels).
>
> **Triplet Loss Calculation**:
>
> **4. Emotion-Based Sample Selection.** For each feature in the batch calculated in step 1, we first identify the samples that belong to the same emotion category using the emotion category label (labels[i]). These samples are grouped together, excluding the current sample (anchor). Then, within this group of samples, we distinguish between the positive samples (those with the same emotion intensity) and the negative samples (those with different emotion intensity). This step ensures that the positive and negative samples are selected from the same emotion.
>
> **5. Generating Triplets.** For each sample (anchor), we form triplets by selecting:
> + Anchor (a): The feature vector of the current sample.
> + Positive (b): A feature vector from samples with the same emotion category and intensity.
> + Negative (c): A feature vector from samples with the same emotion category but different intensity.
>
> **6. Loss Calculation.** For each triplet, we calculate the standard Triplet Loss using the formula:
>
> $L = \max (d(a,p) - d(a,n) + \alpha ,0),d(a,p) = ||{z^a} - {z^p}||_2^2,d(a,n) = ||{z^a} - {z^n}||_2^2$
>
> **7. Per Emotion Calculation.** The triplet loss **for each sample** is calculated as described above, but it is done per emotion because we specifically select positive and negative samples based on both the emotion category and emotion intensity. As a result, the loss is computed in a manner that respects the emotional structure of the data. We compute the total loss for the batch by summing the losses from all the triplets and then averaging them over the batch size.
>
> We hope that this explanation clarifies how the Contrastive Loss and Triplet Loss are computed and how they contribute to emotion-based learning in our model.

---

> > ### Author Response · Authors · 2024-11-24
> >
> > >Q6: L219: What kinds of augmentations were applied to the video frames and audio samples?
> >
> > A6: The main purpose of the applied augmentations is to enhance the robustness and generalization ability of the model. For the video data, following [1], we applied a perspective transformation technique to simulate different viewpoints. This allows the model to capture emotional details from faces in non-frontal perspectives, improving its ability to handle variations in facial orientation. For the audio data, since the RAVDESS dataset was recorded in a quiet environment, we followed [2] in adding noise to the audio samples. This augmentation simulates a more realistic, "wild" audio environment, enabling the model to better generalize to real-world conditions where background noise is often present.
> >
> > [1] Dong J, Wang X, Zhang L, et al. Feature re-learning with data augmentation for video relevance prediction[J]. IEEE Transactions on Knowledge and Data Engineering, 2019, 33(5): 1946-1959.
> >
> > [2] Han T, Huang H, Yang Z, et al. Supervised contrastive learning for accented speech recognition[J]. arXiv preprint arXiv:2107.00921, 2021.
> >
> >
> > >Q7: L225: The latent representations are averaged over the sequence. How is it ensured that continuous emotional information along the sequence is not lost? Could the authors provide more details on the contrastive loss in Equation 4, and clarify if other methods besides averaging were considered, such as mapping to a lower dimension with a learnable layer?
> >
> > A7: Thank you for your insightful question. We employ the audio-visual loss to ensure that emotional details within the sequence dimension are preserved and the detailed calculation process of the contrastive loss is as described in response A5. To clarify, the emotional input to our EMG module (also the output of EDE) consists of sequence-level feature representations. These representations are averaged over the sequence dimension only during the calculation of contrastive and triplet losses.
> >
> > The goal of the **contrastive loss** is to learn a disentangled emotional space at the emotion type level. While employing a learnable layer could allow for finer-grained learning across the sequence dimension, at this stage, our focus is on constructing a **global emotional space**. Within the same emotional category, audio/video data often exhibit minor variations in emotional expression across different time points. However, these variations typically do not significantly affect the overall emotional representation. Hence, we use **averaging** as a representation of the emotional feature for constructing the disentangled emotional space.
> >
> > It is important to note that our EDE module is explicitly designed to capture sequence-level disentangled emotional representations. While the emotional space construction in the contrastive loss calculation does not consider the sequence dimension, potentially leading to a loss of sequence-specific emotional details, we address this issue by introducing an **audio-visual loss**. This loss leverages the rich emotional details present in the visual modality to enable interaction and exchange of emotional features during audio-visual joint training. By doing so, it takes full advantage of the complementarity between modalities, allowing the model to incorporate finer-grained **emotional details** from the visual modality into the learning of audio emotional features. This approach effectively mitigates the potential weakening or loss of sequence-level emotional details that could arise from averaging during the contrastive loss calculation, further enhancing the expressive power and completeness of emotional features.
> >
> > As shown in **Fig.4(b)** in the paper, when training without the visual-audio loss, we observed the exact problem of emotional detail loss described in your comment. For example, the generated outputs only reflect a static overall emotional expression, lacking finer details such as eyebrow movements. However, after incorporating the visual-audio loss, our results exhibit dynamic details, such as eyebrow motions, akin to the ground truth (GT). Additionally, as demonstrated in the demos on our **webpage**, we encourage the reviewer to focus on regions around the eyes and the nasal muscles. These areas clearly showcase the ability of our model to capture subtle emotional details effectively.
> >
> > We hope this explanation addresses your concerns and provides a clear understanding of the rationale behind our design choices. Please let us know if further clarification is needed.

---

> > > ### Author Response · Authors · 2024-11-24
> > >
> > > >Q8: What was the rationale for training the sync expert independently? How are the FLAME region landmarks obtained—are they input as rendered crops or as 3D vertices?
> > >
> > > A8: Thank you for your question. Below is the clarification for the two points you raised:
> > >
> > > 1. Rationale for training the Sync Expert independently. The decision to train the lip-sync expert independently is motivated by several factors:
> > > + Common practice in top-tier works: Approaches such as Wav2Lip[1] and GeneFace[2] also train independent lip-sync experts. This practice has proven effective in achieving high-quality lip-syncing performance.
> > > + Focus on temporal accuracy: Unlike mesh-based losses used in some related works, we wanted to focus on assessing lip-sync accuracy in the temporal domain, rather than simply relying on vertex-based loss (such as GT_vertex - pred_vertex). Training a dedicated lip-sync expert allows for the use of a strong supervision signal, which ensures that the model learns more effectively.
> > > + Limitations of face reconstruction loss: Pixel-based face reconstruction losses cannot accurately constrain audio-to-mouth-sync because face reconstruction loss is computed over the entire image, whereas the lip region only occupies a small portion (less than 4%) of the image. This makes it difficult to focus on lip detail. Additionally, during face reconstruction training, the lip movements are typically optimized only in the later stages, leading to a lack of supervision in the earlier stages. The LVE loss, which only evaluates lip-sync based on single-frame images, also lacks temporal context and fails to capture the quality of dynamic lip movements. Therefore, the independent sync expert was introduced to improve the accuracy and temporal consistency of lip-syncing.
> > >
> > > 2. Obtaining FLAME region landmarks. The FLAME model, which consists of 5023 points, provides 68 facial landmarks, including the lip region. We utilize the provided 68-point mask from the FLAME website to extract the lip region. Specifically, we use the 20 landmarks corresponding to the mouth region, which gives us a total of 60 3D coordinates (20 × 3 = 60). These coordinates are then used as input to our model for accurate lip synchronization.
> > >
> > > [1]Prajwal K R, Mukhopadhyay R, Namboodiri V P, et al. A lip sync expert is all you need for speech to lip generation in the wild[C]//Proceedings of the 28th ACM international conference on multimedia. 2020: 484-492.
> > >
> > > [2]Ye Z, Jiang Z, Ren Y, et al. Geneface: Generalized and high-fidelity audio-driven 3d talking face synthesis[J]. arXiv preprint arXiv:2301.13430, 2023.
> > >
> > > >Q9: fig 1a: The method name is missing in the caption. & fig 1b: The triangle and circle shapes representing features lack labels indicating what they refer to. & fig 4 and Ablation Studies (pages 9,10): "Ecotalk" is used instead of "EcoFace."
> > >
> > > A9: Thanks for your careful and accurate reading of our paper. We have fixed the errors in the revised version.

---

> > > > ### Comment · Reviewer_AbM6 · 2024-11-28
> > > > **Thank you for your comprehensive response.**
> > > >
> > > > Thank you for your comprehensive response. I appreciate the authors efforts in providing updated evaluations for Q2 and the work involved in addressing it. While most of my questions have been thoroughly addressed, I still have concerns regarding the lack of a quantified explanation in Q1, Q3, and Q4. Additionally for Q4, the lack of a demonstration of the proposed approach (e.g. pivot identity conditioning as suggested by the authors) for speaker-specific generations weakens the impact of this key contribution, a concern I share along with other reviewers. Given these concerns, I will maintain my ratings.

---

> > > > > ### Author Response · Authors · 2024-12-01
> > > > >
> > > > > Thank you for your continued valuable feedback and for carefully reviewing the revised manuscript. We appreciate your thoughtful comments and are glad that most of the concerns have been addressed. We understand that a few questions remain unresolved, and we have conducted further experiments to provide clearer answers.
> > > > >
> > > > > >Quantified explanation in Q1
> > > > >
> > > > > To better evaluate the stability of the model when dealing with anomalous data in the training set, we aim to identify anomalous ground truth data within the RAVDESS and HDTF datasets and compare the generation stability of Faceformer, Codetalker, and our model.
> > > > >
> > > > > + Error calculation. To select anomalous data from the training set, we use the camera parameters estimated by EMOCA to obtain the predicted landmark point locations on the image. Specifically, after driving the FLAME model, we can obtain 68 3D landmark points and project them to the image space using the camera parameters to obtain the predicted landmark point locations on the image. We can then use relevant algorithms (e.g. face_alignment) to estimate the 68 landmark point locations on the GT image. The L2 error is calculated for all landmark points and the maximum value of this error is used as an indicator for identifying anomalies. We define this error as LE (landmark error). The formula is as follows: $LE(gt,pred) = \mathop {\max }\limits_{i = 1,...,68} \sum\limits_{j = 1}^2 {{{(g{t_{i,j}} - pre{d_{i,j}})}^2}}.$
> > > > >
> > > > > + Identifying Anomalous Data in the Training Set. We use the Interquartile Range (IQR) method to detect anomalous data. If the L2 error exceeds the threshold defined by $Q_3+1.5×IQR$, where $Q3$ is the third quartile, we consider that data point to be anomalous.
> > > > >
> > > > > + Comparison of Generation Stability. Using the identified anomalous data, we calculate the LE for Faceformer, Codetalker and our model. As there are no anomalous points in the HDTF dataset, we omit this in the results. The results are as follows. It can be seen that despite the anomalies in the training data, our model is still able to have a more stable generation.
> > > > >
> > > > > ||RAVDESS|
> > > > > |:------:|:------:|
> > > > > |GT|681.759|
> > > > > |FaceFormer|455.312|
> > > > > |CodeTalker|408.980|
> > > > > |Ours|185.125|
> > > > >
> > > > > + Another way  to induce stability. EmoTalk[1] proposes a loss function:  $L_{vel}= \parallel (\hat b_t - \hat b_{t-1}) - (b_t - b_{t-1}) \parallel ^2,$ to smooth the impact of anomalous data during training. We also adopted this approach and trained our model using this loss to observe any improvements when handling anomalous data. The results are as follows. Although the introduction of the smoothing loss yields a slight improvement, the difference is minimal. This suggests that our generative method is inherently robust to extreme cases and helps mitigate the effects of anomalies in the data.
> > > > >
> > > > > ||RAVDESS|
> > > > > |:------:|:------:|
> > > > > |GT|681.759|
> > > > > |Ours|185.125|
> > > > > |Ours w/ $L_{vel}$|178.722|
> > > > >
> > > > > [1]Peng Z, Wu H, Song Z, et al. Emotalk: Speech-driven emotional disentanglement for 3d face animation[C]//Proceedings of the IEEE/CVF International Conference on Computer Vision. 2023: 20687-20697.

---

> ### Author Response · Authors · 2024-12-01
>
> >Quantified explanation in Q3
>
> As mentioned in our previous response, when disentangling the feature spaces of different emotions, the intensity of each emotion is already embedded within these features. Therefore, even if two emotions are similar in terms of arousal, their intensities can still be independently distinguished.
>
> To quantitatively analyze whether there is enough distinguishability between the features of different emotions and different intensity levels, we calculated the Euclidean distance matrix for the feature vectors of each emotion and intensity level. The distance between feature vectors for the same emotion but different intensities is computed using the following formula:
>
> $D(i,j) = \frac{1}{{{n_i}{\rm{ \times }}{n_j}}}\sum\limits_{k = 1}^{{n_i}} {\sum\limits_{m = 1}^{{n_j}} {d(x_i^k,x_j^m)} }. $
>
> We consider seven emotions: calm, happy, surprise, angry, disgust, fear and sad, with each emotion having two intensity levels. Therefore, we have a total of 14 categories, where $i,j \in (0,7{\rm{ \times }}2 = 14]$. For each pair of emotion objects, we calculate the Euclidean distance between all possible pairs of feature vectors from both objects. The average of these pairwise Euclidean distances is then taken as the overall Euclidean distance between the two emotion objects.
>
> ||Calm_level1|Calm_level2|Happy_level1|Happy_level2|Sad_level1|Sad_level2|Angry_level1|Angry_level2|Fear_level1|Fear_level2|Disgust_level1|Disgust_level2|Surprise_level1|Surprise_level2|
> |:------:|:------:|:------:|:------:|:------:|:------:|:------:|:------:|:------:|:------:|:------:|:------:|:------:|:------:|:------:|
> |Calm_level1|**0.235**|**0.361**|1.654|1.785|1.372|1.486|2.076|2.203|2.164|2.201|1.873|1.889|2.289|2.339|
> |Calm_level2|**0.361**|**0.251**|1.749|1.867|1.389|1.493|2.208|2.324|2.233|2.268|1.956|1.973|2.405|2.452|
> |Happy_level1|1.654|1.749|**0.253**|**0.313**|1.349|1.327|1.348|1.388|1.288|1.242|1.502|1.438|1.153|1.165|
> |Happy_level2|1.785|1.867|**0.313**|**0.229**|1.411|1.363|1.372|1.377|1.279|1.211|1.553|1.479|1.183|1.179|
> |Sad_level1|1.372|1.389|1.349|1.411|**0.236**|**0.324**|1.751|1.832|1.135|1.174|1.304|1.309|1.694|1.731|
> |Sad_level2|1.486|1.493|1.327|1.363|**0.324**|**0.259**|1.789|1.849|1.095|1.114|1.349|1.343|1.692|1.721|
> |Angry_level1|2.076|2.208|1.348|1.372|1.751|1.789|**0.214**|**0.313**|1.533|1.483|1.241|1.179|1.301|1.274|
> |Angry_level2|2.203|2.324|1.388|1.377|1.832|1.849|**0.313**|**0.142**|1.567|1.494|1.296|1.222|1.349|1.305|
> |Fear_level1|2.164|2.233|1.288|1.279|1.135|1.095|1.533|1.567|**0.253**|**0.337**|1.528|1.487|1.321|1.325|
> |Fear_level2|2.201|2.268|1.242|1.211|1.174|1.114|1.483|1.494|**0.337**|**0.169**|1.539|1.487|1.313|1.306|
> |Disgust_level1|1.873|1.956|1.502|1.553|1.304|1.349|1.241|1.296|1.528|1.539|**0.227**|**0.306**|1.168|1.182|
> |Disgust_level2|1.889|1.973|1.438|1.479|1.309|1.343|1.179|1.222|1.487|1.487|**0.306**|**0.255**|1.122|1.129|
> |Surprise_level1|2.289|2.405|1.153|1.183|1.694|1.692|1.301|1.349|1.321|1.313|1.168|1.122|**0.173**|**0.255**|
> |Surprise_level2|2.339|2.452|1.165|1.179|1.731|1.721|1.274|1.305|1.325|1.306|1.182|1.129|**0.255**|**0.196**|
>
> As can be seen from the results, features of different intensity levels within the same emotion are close but not overly similar, allowing for a certain degree of distinction. On the other hand, the features of different emotions and different intensity levels are far apart. For instance, even in cases like "surprise" and "angry," the features for both the emotion category and the intensity level category show separability.

---

> ### Author Response · Authors · 2024-12-01
>
> >Quantified explanation in Q4
>
> After training with the pivot identity, the performance metrics under the condition of using the pivot identity are as follows.
>
> |||RAVDESS||||HDTF|||VOCASET||||MEAD||
> |:------:|:------:|:------:|:------:|:------:|:------:|:------:|:------:|:------:|:------:|:------:|:------:|:------:|:------:|:------:|
> |Methods|VE-FID↓|LVE↓|LSE-D↓|LSE-C↑|LVE↓|LSE-D↓|LSE-C↑|LVE↓|LSE-D↓|LSE-C↑|VE-FID↓|LVE↓|LSE-D↓|LSE-C↑|
> |Ours|21.57|2.19|9.616|1.010|2.61|10.253|0.823|3.86|10.757|0.743|32.44|5.21|9.113|0.709|
> |Ours (w/ pivot identity)|21.53|2.11|9.601|1.123|2.48|10.167|0.833|3.65|10.698|0.782|33.78|5.73|9.132|0.721|
>
> As mentioned earlier, the use of a pivot identity allows us to quickly fine-tune our decoder for an unseen person, enabling us to capture the specific speaking style of that person efficiently. For example, when testing on an unseen person, we use Actor 24 in RAVDESS as a case study. The results before and after the fine-tuning (within 20 minutes) are as follows.
>
> |Actor-24|VE-FID↓|LVE↓|LSE-D↓|LSE-C↑|
> |:------:|:------:|:------:|:------:|:------:|
> |Pivot identity|23.13|2.32|9.685|1.110|
> |Fine-tune(Speaker-specific)|20.55|2.21|9.695|1.088|
>
> We provide a brief demonstration in [here](https://anonymous.4open.science/r/EcoFace-7A5F/demo/speaker-specific-demo.mp4), where you can observe that after a short fine-tuning period, we are able to capture the speaker's eye expression features, while maintaining the overall emotional expression and the accuracy of lip-sync.
>
> I hope my clarification helps to address the concern. Once again, thank you for your time and consideration. If there are any additional questions or if further details are needed, we would be more than happy to provide further explanations and address any remaining concerns.

---

> ### Author Response · Authors · 2024-12-02
> **Hoping that our response could address your concern**
>
> Dear Reviewer AbM6,
>
> Thank you again for your time and effort in reviewing our work! We would appreciate it if you can let us know if our response has addressed your concern. As the end of the rebuttal phase is approaching, we look forward to hearing from you and remain at your disposal for any further clarification that you might require.
>
> Thanks in advance,
>
> Paper 10320 authors

---

> > ### Comment · Reviewer_AbM6 · 2024-12-03
> > **Rating updated**
> >
> > Thank you again for the elaborative discussion and for addressing my concerns. I have increased my rating to 6. Kindly ensure the updated material is incorporated into the paper.

---

> > > ### Author Response · Authors · 2024-12-03
> > > **Thanks for your acknowledgment of our rebuttal**
> > >
> > > Dear Reviewer,
> > >
> > > Thank you for your positive feedback and for increasing the rating! I greatly appreciate your recognition and will ensure the updated material is carefully incorporated into the paper as suggested. Once again, thank you for your time and thoughtful suggestions!

---

### Official Review · Reviewer_r7xK · 2024-11-03

**Soundness:** 2
**Presentation:** 3
**Contribution:** 3
**Rating:** 6
**Confidence:** 4

**Summary:**

The paper proposes EcoFace, a framework for generating 3D talking faces using speech signals. The framework first constructs an audiovisual emotion space that is independent of the speech content. Using a Variational Autoencoder the framework then generates FLAME parameters (a low-dimensional representation of facial movements). The model’s encoder is conditioned on both speech and emotion features to create a latent representation that captures expressive nuances. Finally, a decoder that is conditioned on the speaker is used for the facial animation generation. Experimental results show better performance in both emotional expressiveness and lip-sync accuracy when compared with state-of-the-art methods.

**Strengths:**

1. Novel approach for emotion disentanglement using speech and visual information.
2. Comprehensive evaluation through quantitative/qualitative metrics, user studies, and ablation experiments.

**Weaknesses:**

1. Some methodological choices are unclear, and additional evaluation details are needed (see questions).
2. Speaker-specific modeling. Generation can only be performed for speakers the model has seen during training.

**Questions:**

1. You mention in the introduction that your model can discriminate the signal of different types and intensities of emotion features. How does your model interpret varying intensities of emotion and how does the contrastive-triplet loss contribute to this?

2. Sec. 4.1: To evaluate the methodology on the test set for unseen subjects you condition on all training identities. Why use all identities rather than a subset?

3. Although you retrained FaceFormer and CodeTalker on RAVDESS and HDTF, you did not retrain EMOTE which seems to provide better results than your approach on MEAD that the model was trained on. For fair comparison could you perform this experiment by retraining EMOTE on the RAVDESS too?

4. Table 3: Can you show the results on each dataset separately and for each emotion? Since your model was trained on RAVDESS it may contain some bias towards that dataset when compared with EMOTE.

5. Sec. 4.3,”Effect of emotion embeddings”: Have you tried to extract content features from an emotional speech signals? How would the model perform in this more challenging experiment?

---

> ### Author Response · Authors · 2024-11-24
>
> We are grateful for your positive review and valuable comments, and we hope our response fully resolves your concerns.
> >Q1: Speaker-specific modeling. Generation can only be performed for speakers the model has seen during training.
>
> A1:Thank you for pointing out this insightful weakness. We greatly appreciate your suggestion and agree that the ability to generalize to unseen speakers is an important area for improvement.
>
> In our current work, we focus on emotionally expressive talking face generation, where a key challenge lies in the fact that emotional expressions vary significantly across individuals. For instance, during expressions of happiness, the degree of zygomatic muscle activation (e.g., smiling amplitude) can differ between individuals, leading to varied smile dynamics. The speaker-specific nuances are critical for generating realistic and emotionally expressive talking faces. As such, our current design emphasizes speaker-specific modeling, allowing the system to capture and reproduce these fine-grained variations. While this approach achieves strong performance for known speakers, we acknowledge that it limits the generalization to unseen subjects.
>
> To address this concern, we explored a generalization strategy proposed in the UniTalker[1]. Specifically, we introduced a **pivot identity** into our model, which acts as a generalized speaker label. During training, we replace the ground truth identity label with this pivot identity label with a probability of 10%. This augmentation encourages the model to learn a generalized representation that is not strictly tied to individual speaker characteristics. We retrained our model and performed quantitative analyses on the test set with pivot identity as a condition, and the results are tabulated below.
>
> |||RAVDESS||||HDTF|||VOCASET||||MEAD||
> |:------:|:------:|:------:|:------:|:------:|:------:|:------:|:------:|:------:|:------:|:------:|:------:|:------:|:------:|:------:|
> |Methods|VE-FID↓|LVE↓|LSE-D↓|LSE-C↑|LVE↓|LSE-D↓|LSE-C↑|LVE↓|LSE-D↓|LSE-C↑|VE-FID↓|LVE↓|LSE-D↓|LSE-C↑|
> |Ours|21.57|2.19|9.616|1.010|2.61|10.253|0.823|3.86|10.757|0.743|32.44|5.21|9.113|0.709|
> |Ours (w pivot identity)|21.53|2.11|9.601|1.123|2.48|10.167|0.833|3.65|10.698|0.782|33.78|5.73|9.132|0.721|
>
> We observed that similar results were achieved using the pivot identity. Notably, for unseen subjects, we can use the pivot identity as a condition to **quickly fine-tune our EMG decoder, adapting it to the speaking style of new individuals**. This approach helps improve the model's generalization capability.
>
> We appreciate your feedback, which motivated us to explore this improvement. We believe this addition strengthens the contribution of our work and opens up exciting future directions, such as further balancing generalization and speaker-specific detail preservation.
>
> [1]Fan X, Li J, Lin Z, et al. UniTalker: Scaling up Audio-Driven 3D Facial Animation through A Unified Model[J]. arXiv preprint arXiv:2408.00762, 2024.

---

> > ### Author Response · Authors · 2024-11-24
> >
> > >Q2: You mention in the introduction that your model can discriminate the signal of different types and intensities of emotion features. How does your model interpret varying intensities of emotion and how does the contrastive-triplet loss contribute to this?
> >
> > A2: In our model, the contrastive-triplet loss is a combination of two distinct components: the contrastive loss and the triplet loss, which are defined in equations (4) and (7) of the paper, respectively. Specifically, the contrastive loss minimizes the distance between latent features of the same emotion while maximizing the distance between different emotions based on the ground-truth emotion labels. This encourages the model to learn well-separated emotion-specific latent spaces. The triplet loss, on the other hand, operates in a single emotion space with the ground-truth intensity labels (which are classified into two categories: 0 for lower intensity and 1 for higher intensity). For each cluster of features corresponding to the same emotion, the triplet loss seeks to enforce a finer distinction in the intensity by comparing features of different intensities. In detail, for each emotion feature in the batch, we treat it as an anchor and then search for positive examples (i.e., features from the same emotion with the same intensity) and negative examples (i.e., features from the same emotion but with different intensities). The triplet loss aims to maximize the distance between the anchor and the negative example, while ensuring that the distance between the anchor and positive examples remains smaller than the difference by a margin defined by a threshold. This process facilitates the differentiation of emotional intensities by pushing features of different intensities apart while keeping features of the same intensity closer together.
> >
> > Our motivation for combining the contrastive and triplet losses to model emotional intensity is informed by the well-established Valence-Arousal (VA) model, a widely used framework in emotion research. The VA model suggests that emotions can be represented as combinations of valence and arousal, with arousal being particularly key in distinguishing between different emotional intensities. This implies that intensity assessment is inherently tied to the emotional type assessment (via arousal), where emotions of the same type share a similar valence but can differ subtly in arousal. Therefore, our approach first allows the model to learn separate latent spaces for each emotion, effectively distinguishing the arousal and valence ranges for different emotions via the contrastive loss. Afterward, the triplet loss works within each emotion's latent space to further differentiate emotions based on their intensity by focusing on the subtle differences in arousal. This two-step process ensures that our model can effectively discriminate between both emotional types and their varying intensities.
> >
> > We have included visualizations and a more detailed analysis of the intensity differentiation in the **Appendix B** of the paper. We hope this explanation clarifies the contribution of the contrastive-triplet loss in modeling emotional intensity, and we appreciate your further review of the updated manuscript.

---

> > > ### Author Response · Authors · 2024-11-24
> > >
> > > >Q3: Sec. 4.1: To evaluate the methodology on the test set for unseen subjects you condition on all training identities. Why use all identities rather than a subset?
> > >
> > > A3: We suspect that there might have been a minor misinterpretation regarding our approach to conditioning on all training identities for unseen characters in the test set. Allow us to clarify the methodology and the motivation behind this choice.
> > >
> > > 1. Explanation of Conditioning on All Training Identities. In our experiments, the training datasets (RAVDESS and HDTF) include 24 and 131 identities, respectively. We selected 90% of the characters as training identities (140) and 10% as test identities (unseen). And we select 10% randomly from the data of training identites to be used as validation set and the rest as training set. During the quantitative evaluation, for each speech input, we iteratively conditioned the model on all 140 training identities as the style condition, generating 140 results for each input. The quantitative metrics were calculated for each generated result, and the final score was obtained by averaging across all these results.
> > >
> > > 2. Adherence to Common Practice. This evaluation strategy follows conventions used in prior works such as FaceFormer[1] and CodeTalker[2], which also condition on all training identities to comprehensively assess model performance. This ensures consistency with the evaluation methodologies of state-of-the-art studies.
> > >
> > > 3. Motivation for Conditioning on All Identities. Using all training identities for evaluation provides a more holistic perspective on the model’s generalization and stability. While using a subset—or even a single identity—could also reflect model performance (even better quantitative results, such as the results in Q1 by using the pivot identity.), it may not adequately capture potential overfitting to specific styles. Overfitting could result in issues like misaligned lip synchronization, distorted facial expressions, or inappropriate emotional displays. Furthermore, although speaking styles may vary, the fundamental emotional expressions (e.g., happiness, sadness) are generally consistent across individuals apart from minor stylistic nuances. Similarly, lip synchronization should remain unaffected by the choice of identity. By conditioning on all identities, we can observe if the model’s performance is robust across diverse styles and does not exhibit identity-specific artifacts or biases.
> > >
> > > [1]Fan Y, Lin Z, Saito J, et al. Faceformer: Speech-driven 3d facial animation with transformers[C]//Proceedings of the IEEE/CVF Conference on Computer Vision and Pattern Recognition. 2022: 18770-18780.
> > >
> > > [2]Xing J, Xia M, Zhang Y, et al. Codetalker: Speech-driven 3d facial animation with discrete motion prior[C]//Proceedings of the IEEE/CVF Conference on Computer Vision and Pattern Recognition. 2023: 12780-12790.

---

> ### Author Response · Authors · 2024-11-24
>
> >Q4: Although you retrained FaceFormer and CodeTalker on RAVDESS and HDTF, you did not retrain EMOTE which seems to provide better results than your approach on MEAD that the model was trained on. For fair comparison could you perform this experiment by retraining EMOTE on the RAVDESS too?
>
> A4: Thank you for highlighting this important point. We appreciate the opportunity to clarify our experimental choices and to address your suggestion.
> Initially, we chose not to retrain EMOTE because our aim was to evaluate the generalization capability of our model, which was trained on the RAVDESS dataset, to a new dataset like MEAD. Both RAVDESS and MEAD are emotion-rich datasets featuring audio-visual recordings of actors expressing different emotions. By not retraining EMOTE, we intended to measure how well our model could adapt to a dataset it had not been trained on, compared to EMOTE’s performance on the dataset it was specifically trained on (MEAD).
>
> After receiving your valuable feedback, we revisited this aspect to better understand why our model underperforms on MEAD compared to EMOTE. We observed that, due to the influence of speaker styles, some test samples in the MEAD dataset exhibit relatively subdued emotional expressions in speech. This weakens the performance of our model, which relies on extracting emotional features from speech. In contrast, EMOTE directly specifies emotions through labels for generation, making it immune to such effects.
>
> In response to your suggestion, we have retrained EMOTE on both the RAVDESS and HDTF datasets. The results of this experiment are presented in the table below.
>
> |||RAVDESS||||HDTF|||VOCASET||||MEAD||
> |:------:|:------:|:------:|:------:|:------:|:------:|:------:|:------:|:------:|:------:|:------:|:------:|:------:|:------:|:------:|
> |Methods|VE-FID↓|LVE↓|LSE-D↓|LSE-C↑|LVE↓|LSE-D↓|LSE-C↑|LVE↓|LSE-D↓|LSE-C↑|VE-FID↓|LVE↓|LSE-D↓|LSE-C↑|
> |Emote (retrained)|27.01|3.65|10.152|0.914|3.61|10.967|0.801|4.01|10.911|0.688|28.96|6.02|10.010|0.703|
> |Ours|21.57|2.19|9.616|1.010|2.61|10.253|0.823|3.86|10.757|0.743|32.44|5.21|9.113|0.709|
>
> We acknowledge that EMOTE’s use of explicit labels provides a more direct mapping for emotion-driven generation. However, our approach focuses on learning nuanced emotional details from speech sequences, which enhances the granularity of emotional dynamics. These strengths are reflected in the superior performance of our model on RAVDESS.
>
> >Q5: Table 3: Can you show the results on each dataset separately and for each emotion? Since your model was trained on RAVDESS it may contain some bias towards that dataset when compared with EMOTE.
>
> A5: Thank you for pointing this out. We have now provided the results for each dataset and emotion separately.
>
> Here are the results of the User study on the Vocaset：
>
> |Methods|Full-face|Lip-sync|Emotion expression|
> |:------:|:------:|:------:|:------:|
> |Faceformer|16.24%|14.57%|6.16%|
> |CodeTalker|14.03%|12.97%|5.02%|
> |Emoete|8.47%|14.28%|20.69%|
> |EmoTalk|15.31%|14.14%|11.56%|
> |UniTalker|17.98%|21.86%|13.49%|
> |Ours|27.97%|22.18%|43.08%|
>
> Here are the results of the User study on the RAVDESS：
>
> |Methods|Full-face|Lip-sync|Emotion expression|
> |:------:|:------:|:------:|:------:|
> |Faceformer|9.52%|12.53%|7.36%|
> |CodeTalker|9.09%|10.85%|5.48%|
> |Emoete|11.21%|12.79%|21.03%|
> |EmoTalk|15.92%|12.01%|12.27%|
> |UniTalker|21.52%|18.09%|13.74%|
> |Ours|32.74%|33.73%|40.12%|
>
> Here are the results of the User study for each emotion on the RAVDESS:
>
> |Methods|Happy|Surprised|Sad|Angry|Fearful|Disgust|
> |:------:|:------:|:------:|:------:|:------:|:------:|:------:|
> |Faceformer|7.31%|6.22%|9.09%|8.08%|9.00%|4.21%|
> |CodeTalker|5.85%|6.69%|5.45%|4.05%|5.00%|5.78%|
> |Emoete|21.95%|20.57%|20.91%|22.22%|19.50%|21.05%|
> |EmoTalk|10.24%|11.96%|13.18%|11.11%|12.50%|14.73%|
> |UniTalker|15.12%|12.91%|12.72%|11.11%|14.50%|16.31%|
> |Ours|39.53%|41.65%|38.65%|43.43%|39.50%|37.92%|
>
> To further address the potential bias, we conducted the same user study comparing the retrained EMOTE model with our original model on the RAVDESS dataset. The results of this additional user study are presented below.
>
> |Methods|Full-face|Lip-sync|Emotion expression|
> |:------:|:------:|:------:|:------:|
> |Emote (retrained)|37.67%|42.15%|32.88%|
> |Ours|62.33%|57.85%|67.12%|
>
> As shown in the table, our model achieves better performance in both the original and retrained emote. This highlights the generalizability and effectiveness of our approach, despite differences in training datasets.

---

> ### Author Response · Authors · 2024-11-24
>
> >Q6: Sec. 4.3,”Effect of emotion embeddings”: Have you tried to extract content features from an emotional speech signals? How would the model perform in this more challenging experiment?
>
> A6: Thank you for raising this important question. In our original setup, we chose to extract emotion features from neutral speech because Hubert demonstrates a high similarity in content features extracted from speech signals with identical content but varying emotions.
>
> To evaluate the model's performance under this more challenging scenario, we conducted an additional experiment where content features were extracted from emotional speech signals rather than neutral speech. Specifically, we selected one speech sample from each of the seven emotional categories (excluding neutral) to extract the text content. For the emotion features, we used speech and video samples corresponding to the remaining emotions and combined these with the extracted text content as input to generate results. The generated outputs were then evaluated using LSE-C, LSE-D, and VE-FID metrics. The results are presented in the table below.
>
> |Emotional features extracted from|LSE-C↑|LSE-D↓|VE-FID↓|
> |:------:|:------:|:------:|:------:|
> |Audio|1.075|9.899|26.54|
> |Video|1.101|9.983|28.13|
> |GT|1.103|9.977|/|
>
> As shown, even when extracting content from emotional speech signals and subsequently performing emotion swapping for generation, our model maintained robust performance across the evaluation metrics. This demonstrates the resilience and adaptability of our approach in handling emotionally infused inputs.
>
> Furthermore, we have updated the qualitative results for this experiment in the **appendix** of the revised paper and on our **webpage**. We hope these supplementary experiments address your concerns and enrich your understanding of our model's capabilities.

---

> ### Comment · Reviewer_r7xK · 2024-11-28
> **Review Update: Response to Rebuttal**
>
> I would like to thank the authors for thoroughly addressing my concerns and providing the requested additional results. Based on this I have decided to update my rating from 5 to 6.

---

> ### Author Response · Authors · 2024-11-28
> **Thanks for your acknowledgment of our rebuttal**
>
> Dear Reviewer,
>
> Thank you very much for updating your rating based on the additional results and revisions we provided! We greatly appreciate your recognition of the improvements made to the paper, and your feedback has been invaluable in enhancing the quality of our work. Once again, thank you for your time and thoughtful suggestions!

---

### Meta-Review · Area_Chair_2fL4 · 2024-12-23

**Metareview:**

The submission proposes EcoFace, a method for generating 3D talking heads from input speech. To overcome issues exhibited by prior work, the authors introduce explicit signals for emotional motion representation and intensity control using audio-visual loss and a contrastive triplet loss to distinguish emotion intensities.
The submission received final ratings of 6, 6, 6, 8.
The ACs did not find enough reason to overturn the positive consensus and recommend acceptance.

**Additional Comments On Reviewer Discussion:**

Some key weaknesses pointed out by the reviewers include:
- Lack of generalization to unseen speakers
- Lack of details and comparisons

The authors provided detailed answers to all the concerns, and the reviewers ended up increasing the rating given to the submission.

---

### Decision · Program_Chairs · 2025-01-22

Accept (Poster)